# Transcriptional Regulation of Plant Biomass Degradation and Carbohydrate Utilization Genes in the Extreme Thermophile *Caldicellulosiruptor bescii*

Dmitry A. Rodionov,[a,b] Irina A. Rodionova,[c] Vladimir A. Rodionov,[b] Aleksandr A. Arzamasov,[a,b] Ke Zhang,[d] Gabriel M. Rubinstein,[e] Tania N. N. Tanwee,[e] Ryan G. Bing,[f] James R. Crosby,[f] Intawat Nookaew,[g,i] Mirko Basen,[h] Steven D. Brown,[i*] Charlotte M. Wilson,[i,j] Dawn M. Klingeman,[i] Farris L. Poole II,[e] Ying Zhang,[d] Robert M. Kelly,[f] Michael W. W. Adams[e]

[a]Sanford-Burnhams-Prebys Medical Discovery Institute, La Jolla, California, USA
[b]A.A. Kharkevich Institute for Information Transmission Problems, Russian Academy of Sciences, Moscow, Russia
[c]Department of Bioengineering, University of California—San Diego, La Jolla, California, USA
[d]Department of Cell and Molecular Biology, College of the Environment and Life Sciences, University of Rhode Island, Kingston, Rhode Island, USA
[e]Department of Biochemistry and Molecular Biology, University of Georgia, Athens, Georgia, USA
[f]Department of Chemical and Biomolecular Engineering, North Carolina State University, Raleigh, North Carolina, USA
[g]Department of Biomedical Informatics, College of Medicine, University of Arkansas for Medical Sciences, Little Rock, Arkansas, USA
[h]Mathematisch-Naturwissenschaftliche Fakultät, Institut für Biowissenschaften, Mikrobiologie, Universität Rostock, Rostock, Germany
[i]Biosciences Division, Oak Ridge National Laboratory, Oak Ridge, Tennessee, USA
[j]University of Otago, Dunedin, New Zealand

**ABSTRACT** Extremely thermophilic bacteria from the genus *Caldicellulosiruptor* can degrade polysaccharide components of plant cell walls and subsequently utilize the constituting mono- and oligosaccharides. Through metabolic engineering, ethanol and other industrially important end products can be produced. Previous experimental studies identified a variety of carbohydrate-active enzymes in model species *Caldicellulosiruptor saccharolyticus* and *Caldicellulosiruptor bescii*, while prior transcriptomic experiments identified their putative carbohydrate uptake transporters. We investigated the mechanisms of transcriptional regulation of carbohydrate utilization genes using a comparative genomics approach applied to 14 *Caldicellulosiruptor* species. The reconstruction of carbohydrate utilization regulatory network includes the predicted binding sites for 34 mostly local regulators and point to the regulatory mechanisms controlling expression of genes involved in degradation of plant biomass. The Rex and CggR regulons control the central glycolytic and primary redox reactions. The identified transcription factor binding sites and regulons were validated with transcriptomic and transcription start site experimental data for *C. bescii* grown on cellulose, cellobiose, glucose, xylan, and xylose. The XylR and XynR regulons control xylan-induced transcriptional response of genes involved in degradation of xylan and xylose utilization. The reconstructed regulons informed the carbohydrate utilization reconstruction analysis and improved functional annotations of 51 transporters and 11 catabolic enzymes. Using gene deletion, we confirmed that the shared ATPase component MsmK is essential for growth on oligo- and polysaccharides but not for the utilization of monosaccharides. By elucidating the carbohydrate utilization framework in *C. bescii*, strategies for metabolic engineering can be pursued to optimize yields of bio-based fuels and chemicals from lignocellulose.

**IMPORTANCE** To develop functional metabolic engineering platforms for nonmodel microorganisms, a comprehensive understanding of the physiological and metabolic characteristics is critical. *Caldicellulosiruptor bescii* and other species in this genus have untapped potential for conversion of unpretreated plant biomass into industrial fuels and chemicals. The highly interactive and complex machinery used by *C. bescii*

Address correspondence to Dmitry A. Rodionov, rodionov@sbpdiscovery.org, or Michael W. W. Adams, adamsm@uga.edu.

* Present address: Steven D. Brown, LanzaTech, Inc., Skokie, Illinois, USA.

For a companion article on this topic, see https://doi.org/10.1128/mSystems.01351-20.

to acquire and process complex carbohydrates contained in lignocellulose was elucidated here to complement related efforts to develop a metabolic engineering platform with this bacterium. Guided by the findings here, a clearer picture of how *C. bescii* natively drives carbohydrate utilization is provided and strategies to engineer this bacterium for optimal conversion of lignocellulose to commercial products emerge.

**KEYWORDS** *Caldicellulosiruptor bescii*, lignocellulose degradation, carbohydrate metabolism, transcriptional regulation, *Caldicellulosiruptor*, carbohydrate utilization, comparative genomics, metabolic reconstruction, plant biomass degradation, regulon

Plant biomass contains carbohydrate-rich cell walls whose composition is highly diverse and differs significantly between plants. A limited number of monosaccharides comprise the majority of plant polysaccharides (1). D-Glucose is the major component of glucans, cellulose in particular, while D-xylose is the predominant component of hemicelluloses, such as xylan, glucuronoxylan, arabinoxylan, and glucuronoarabinoxylan. The latter four hemicelluloses also contain residues of L-arabinose, D-glucuronate, and more rarely, D-galactose. Mannans (glucomannan, galactomannan, and galactoglucomannan) are composed of D-mannose, D-glucose, and D-galactose (2). Pectins (homogalacturonans and rhamnogalacturonans), acting as binders between cellulose and hemicelluloses fibers, are composed of D-galacturonate and L-rhamnose residues with various branching side chains (3). Starches (amylose and amylopectin) function as glucose storage polysaccharides in many plants and contain linear or branching D-glucose chains with either $\alpha$-1,4 or $\alpha$-1,6 glycosidic linkages, often referred to as $\alpha$-glucans.

Microbial conversion of plant biomass into biofuels involves extensive and highly diversified carbohydrate utilization (CU) pathways that usually involve three major components. First, degradation of plant cell wall polysaccharides requires an array of extracellular carbohydrate-active enzymes (CAZymes), including glycosyl hydrolases (GHs) with carbohydrate-binding modules (CBMs), and polysaccharide lyases (PLs). Second, the extracellular biomass-derived oligo- and monosaccharides are transported into the cytoplasm via specific uptake transporters. Third, the imported saccharides undergo intracellular processing via sets of enzymes organized into individual catabolic pathways. These ultimately yield central metabolites that are further degraded through major glycolytic pathways (glycolysis or pentose-phosphate pathway) and natively fermented to final products (e.g., acetate, lactate, or ethanol).

Extremely thermophilic, cellulolytic bacteria from the genus *Caldicellulosiruptor* possess extensive and highly diversified machinery for plant biomass degradation and carbohydrate utilization (4). *Caldicellulosiruptor* species are common in neutral pH terrestrial hot springs globally where they are the dominant inhabitants of these high-temperature (70 to 80°C) environments and are capable of growth on cellulose, xylan, pectin, and other complex glucan substrates (5). Recent availability of genomic sequences of 14 *Caldicellulosiruptor* species provides opportunities for comparative analyses of inventories of CAZymes and CU genes, thereby linking them to species with distinct physiological properties. The glucan degradation locus (GDL) in *C. bescii* encodes six multidomain GHs/CBMs (multifunctional cellulases), three PLs (pectate lyases and rhamnogalacturonan lyase), and several other cellulose-binding proteins. This arsenal of extracellular proteins participates in primary breakdown of a variety of complex polysaccharides, including cellulose, $\beta$-glucan, xylan, mannan, and pectin (6, 7). Other cellulolytic *Caldicellulosiruptor* species possess GDLs with highly variable GH/CBM domain contents, suggesting dynamic evolutionary adaptation to variable plant biomass composition (8). Despite a diversity of CAZymes found in the genus, representing at least 45 GH and 20 CBM protein families, only some of these proteins have been characterized (5). Also, many metabolic gaps remain in describing CU pathways in *Caldicellulosiruptor* spp., and the identity of many mono- and oligosaccharide transporters is still unknown.

Transcriptomic analysis provides an opportunity to identify missing enzymes and transporters of CU pathways in *Caldicellulosiruptor* species. Previous transcriptomic studies in *C. bescii*, *C. saccharolyticus*, and *C. kronotskyensis* identified groups of genes encoding GHs/CBMs and ABC transporters that are upregulated on plant biomass substrates cellulose and switchgrass (9). Furthermore, the transcriptional response of *C. saccharolyticus* to various monosaccharides (10) and polysaccharides, including cellulose, mannan, pectin, and xylan (11), revealed patterns of genes that are coordinately regulated in response to specific carbohydrates. However, the specific transcription factors (TFs) and regulatory mechanisms for the CU gene network in *Caldicellulosiruptor* species remain unexplored. The global redox-sensing regulator Rex is the only experimentally characterized TF; it was shown to regulate genes encoding a membrane-bound [NiFe]-hydrogenase (MBH), and cytosolic electron-bifurcating [FeFe]-hydrogenase (BF-H$_2$ase) and pyruvate ferredoxin oxidoreductase (POR) in *C. bescii* and *C. saccharolyticus* (12, 13).

Accurate functional assignment of CU genes across diverse bacteria with complete genomes is challenging due to frequent variations of the respective pathways (e.g., nonorthologous gene displacements and functionally divergent paralogs). To enable accurate functional assignment of CU genes and inference of their transcriptional regulatory networks, we previously combined a subsystems-based comparative genomic approach (14, 15) with available experimental data to reconstruct CU metabolic pathways and transcriptional regulons across multiple taxonomic groups of bacteria, including the genera *Thermotoga*, *Shewanella*, *Bacteroides*, and *Bifidobacterium* (16–20), in addition to microbial communities (21). This approach substantially improved the accuracy of genomic annotations and the predicted functions of many previously uncharacterized CU genes in *Thermotoga maritima* (22–25), *Shewanella* spp. (26–29), and other bacterial species (30).

Here, this comparative genomics approach was used to reconstruct CU metabolic and regulatory networks in 14 available genomes from the genus *Caldicellulosiruptor*, with a specific emphasis on *C. bescii*. Comparative analysis of upstream regions of CU genes identified candidate DNA-binding motifs and enabled reconstruction of regulons for numerous TFs. Bioinformatics analyses complemented transcriptomics targeted to reveal transcriptional responses of *C. bescii* to cellulose, cellobiose, glucose, xylan, and xylose, which led to the identification of transcript boundaries for CU genes providing additional validation of the genomic inferences. As result, the reconstructed regulatory network of CU genes refined and improved a metabolic reconstruction describing how *C. bescii* degrades plant biomass and obtains carbon and energy for growth via cytoplasmic CU pathways. The refined CU pathway reconstructions were further used to construct the first metabolic model of *C. bescii* (31), thus opening opportunities for metabolic engineering of this species to produce bio-based chemicals from plant biomass.

## RESULTS AND DISCUSSION

**Genomic reconstruction of carbohydrate utilization pathways and regulons. (i) Glycosyl hydrolases.** To estimate the carbohydrate degradation capabilities of *C. bescii*, sets of CAZymes (GHs and PLs) that are involved in the breakdown of poly- and oligosaccharides into monosaccharides were identified. Overall, 56 proteins containing at least one GH domain were found in the *C. bescii* genome. These include six multidomain proteins containing two to five GH domains that also contain N-terminal signal peptides suggesting their extracellular localization. These secreted GHs are encoded within the glucan degradation locus (GDL), which plays an essential role in plant biomass deconstruction (6). Nine additional GHs, four PLs, and one polysaccharide deacetylase have signal peptides and thus are predicted to function as extracellular enzymes (see Data Set S1A in the supplemental material). Most of these secreted CAZymes (16 of 20) also contain one to three non-catalytic carbohydrate-binding modules (CBMs) that promote association of catalytic domains with polysaccharide. Using similarity searches and metabolic reconstructions via comparative genomics techniques, potential functions of 60 CAZymes that are presumably involved in biomass breakdown and

carbohydrate utilization (CU) in *C. bescii* were examined. As a result, 57 CAZymes were assigned putative substrate specificity, Enzyme Commission (EC) number and CU metabolic pathway (see Data Set S1). These include 5 cytoplasmic and 11 secreted enzymes that were previously characterized in *C. bescii*, *C. saccharolyticus*, or *C. owensensis*. Functional roles in CU for the remaining 41 CAZymes were predicted, based on the genomic context analysis combined with metabolic reconstruction (see below) and by taking into account annotation of experimentally characterized homologs in other organisms (as summarized in Data Set S1A). In summary, the secreted CAZymes are involved in degradation of plant-derived polymers, including cellulose, mannans, hemicelluloses, pectin, and starches, while the cytoplasmic GH enzymes have broader functionalities since they are capable of catabolizing a variety of oligo- and disaccharides to release sugars that enter the central carbohydrate metabolism of *C. bescii*.

(ii) **Carbohydrate utilization pathways.** To identify the repertoire of known and putative CU genes in *C. bescii*, a subsystem-based bioinformatics workflow for the analysis of 14 *Caldicellulosiruptor* genomes was generated. Functional annotation of the identified genes was conducted using genome context analysis, metabolic reconstruction, and homology searches. The complete list of CU genes with their assigned functional roles, CU pathways, protein families, and cellular localization, along with reference information on functionally characterized homologs, are provided in Data Set S1A. The identified CU machinery consists of 222 *C. bescii* genes, including 119 genes encoding catabolic enzymes, 68 genes for 21 carbohydrate transport systems, and 35 genes for DNA-binding transcription factors (TFs) and their cognate sensor kinases (Table 1). Functional annotations of 112 CU genes was supported by published literature, since they were homologous to characterized genes from *Caldicellulosiruptor* or other organisms (see Data Set S1A). Using the subsystem-based workflow, we inferred specific functional assignments for 94 genes whose functions were previously unknown or defined at the level of general class only. Most of these novel functional assignments constitute components of carbohydrate-specific ABC transporters (54 genes) and 29 transcriptional regulators. The ABC transporter systems so identified were classified into two large families, CUT1 and CUT2, that are implicated in uptake of oligo- and monosaccharides, respectively. Eleven novel cytoplasmic enzymes were implicated in the catabolic pathways for galacturonate, arabinose, rhamnose, and fucose (Fig. 1). Seventeen putative genes, including 5 CAZymes, a putative oligosaccharide ABC transporter (Athe_0085-87), a hypothetical CU gene locus (Athe_1993-97), and a putative dehydrogenase in the rhamnose utilization gene locus (Athe_0702-4), have not been assigned a specific function or substrate specificity. Overall, metabolic pathways for the degradation of the major plant polysaccharides, cellulose, xylan, pectin, glucomannan, and $\alpha$-glucans (or starches) were identified, together with the uptake and intracellular catabolism of derived oligo- and monosaccharides (Table 1 and Fig. 1).

(iii) **Carbohydrate utilization regulons.** Bacterial CU-specific regulators tend to cluster on the chromosome with their target genes (16, 18, 21, 32). The identified CU gene loci in *C. bescii* contain 28 genes encoding DNA-binding TFs and 6 genes encoding sensory histidine kinases that are essential components of two-component regulatory systems (TCRS), along with their cognate response regulators (see Data Set S1A). Most of the predicted TFs belong to the protein families whose members were previously known to regulate CU pathways, such as AraC, DeoR, GntR, IclR, LacI, ROK, and RpiR (Fig. 1). In contrast, ArsR constitutes a large family of repressors that collectively provide resistance to a wide range of both biologically required and toxic heavy-metal ions (33). Here, the first example of an ArsR-family TF (named ArbA) involved in CU was found that potentially controls the arabinoside catabolic gene *abfA* in *Caldicellulosiruptor* spp. The majority of identified in *C. bescii* CU-linked TFs lack previously characterized orthologs in any bacterial species and have only homologs with low sequence similarity (below 35 to 40% identity). This suggests that the biological function of these novel TFs (including specificities to effectors and their recognized DNA motifs) cannot be assigned based on homology.

**TABLE 1** Distribution of carbohydrate utilization genes in *C. bescii*[a]

| Sugar utilization pathway | Regulator(s)[b] | Transporter(s)[c] | No. of enzymes (no. secreted) | Total no. of genes | Total no. of genes in regulons |
|---|---|---|---|---|---|
| Monosaccharides | AxgR, GxgRS, AraR* | AxgFGHG1, GxgABCA1X | | 12 | 12 |
| Xylan, xylose | XylR, XynR, XylQ, AxuRS, BxgRS | XynUVW, XloEFG, AxoFGH, AxuABC, BxgLFGL1 | 16 (6) | 39 | 37 |
| Arabinose, arabinose OS | AraR, ArbA, AxeRS | AxeUVW | 5 | 12 | 9 |
| Cellulose, cellobiose | AviRS | AviABCD | 10 (7) | 16 | 12 |
| Pectin, galacturonate | UxaR, PecR, HemR, PecR2, KdgR* | PecXYZX, HemXYZ | 15 (3) | 26 | 20 |
| Glucuronate, glucuronides | KdgR, XynR* | | 10 | 11 | 9 |
| Fucose, fucose OS | FucR | | 3 | 4 | 4 |
| Rhamnose, rhamnose OS, rhamnogalacturonides | RhaR, RhaR2 | RhaXYZ, RqlFGH | 10 (1) | 18 | 8 |
| Galactose, galactose OS | | | 7 | 7 | - |
| Fructose | FruR | FruAB | 3 | 6 | 6 |
| Maltose, maltose OS | MalR, MalR1, MalR2 | MalEFG, MalEFG2 | 9 (3) | 18 | 13 |
| Kojibiose, trehalose | KojR | KojEFG | 3 | 7 | 6 |
| Mannose, mannose OS | MosR, MosSQ | MosABC | 9 | 15 | 7 |
| Deoxynucleosides | DeoR | DeoUVWZ | 6 | 11 | 11 |
| Various oligosaccharides | XylR* | MsmK[c] | | 1 | 1 |
| Other CU genes | IolR | Athe_0085-87 | 10 | 14 | 2 |
| Hypothetical CU pathway oligosaccharides | Athe_1993 | Athe_1997 | 3 | 5 | 5 |
| Total CU genes | 29 (35 genes) | 21 (68 genes) | 119 | 222 | 162 |
| Glycolysis | CggR, Rex* | | 12 | 13 | 6 |
| Central carbon metabolism | Rex | | 10 | 30 | 26 |

[a]The details of all functional assignments summarized here are provided in Data Set S1.

[b]An asterisk indicates a regulator that controls multiple metabolic pathways.

[c]CUT1 family ABC transporters are underlined. MsmK is a common ATPase component shared by multiple of ABC transporters from the CUT1 family.

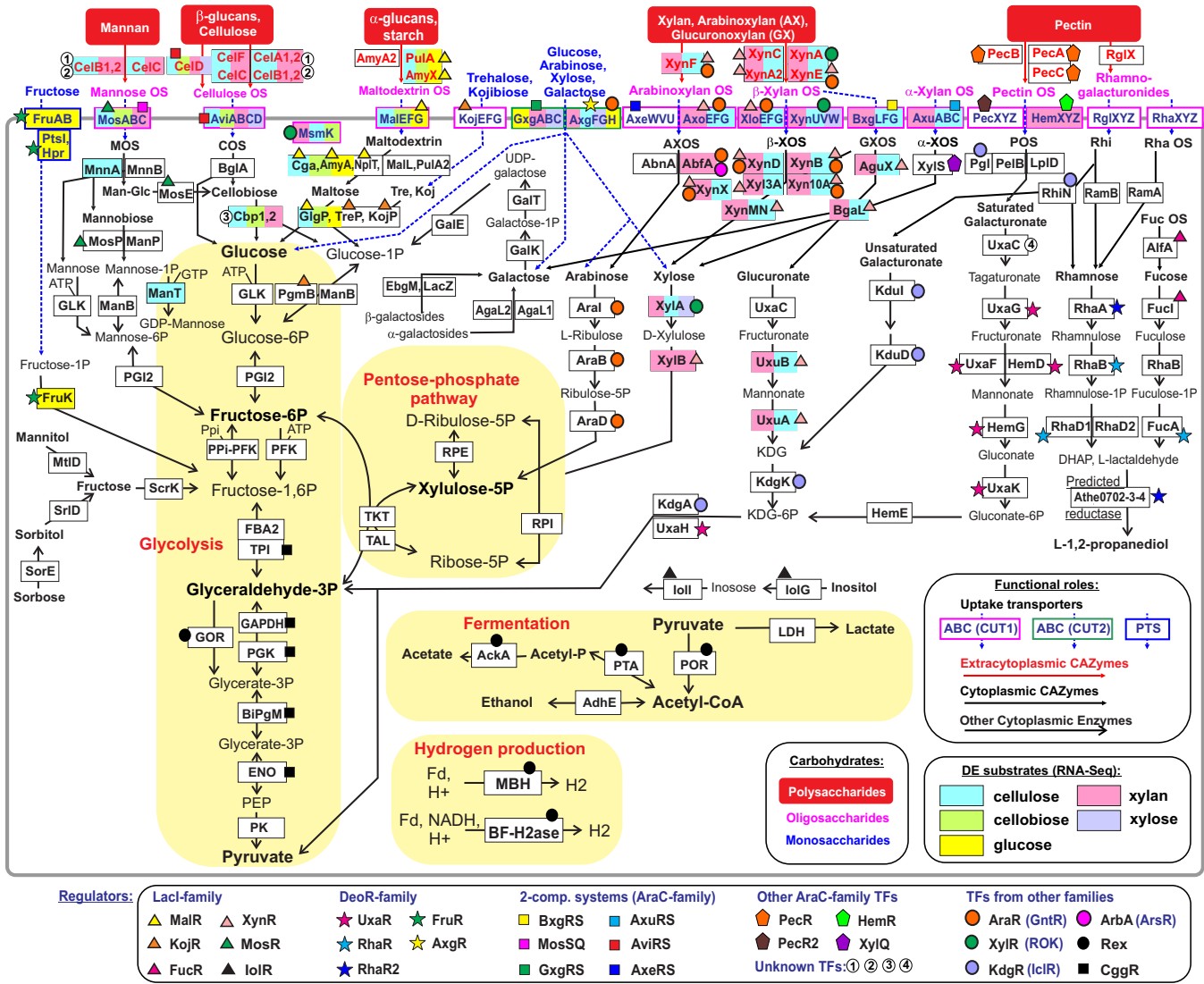

**FIG 1** Reconstruction of metabolic pathways and regulons involved in plant polysaccharide degradation and carbohydrate utilization in *C. bescii*. Polysaccharide substrates are in red bent rectangles. Extracellular enzymes, transporters, and cytoplasmic enzymes are indicated by rectangles with red, blue, and black text, respectively. CAZymes and other classes of enzymes are indicated by large and small arrows, respectively. Novel carbohydrate-specific ABC transporters from the CUT1 and CUT2 families are in boxes with red and green outlines, respectively. The PTS transporter for fructose is in a blue outline box. The genes regulated by the same TF are indicated by matching colored symbols described in the lower inset. The names of differentially expressed (DE) genes on five carbohydrate substrates measured in RNA-Seq experiments are highlighted by light blue (cellulose), green (cellobiose), yellow (glucose), pink (xylan), and pastel blue (xylose). The central carbohydrate metabolism, fermentation, and hydrogen production pathways are indicated by a light yellow background. Detailed information on each gene from the reconstructed CU pathways, including gene IDs, functional roles, etc., is provided in Data Set S1A.

To predict candidate TF binding site (TFBS) motifs for novel regulators, candidate coregulated genes were selected by genome context analysis in all *Caldicellulosiruptor* genomes (see Data Set S2) to which a phylogenetic footprinting approach to upstream regions of these genes was applied (see Fig. S1). After construction of a positional weight matrix for each identified DNA motif, additional TFBSs were sought in each analyzed genome possessing an orthologous TF gene, for which cross-species comparisons of the predicted regulons were done. As a result, DNA-binding motifs were predicted (Fig. 2), and regulons were reconstructed for 26 of 29 CU-related regulators in *C. bescii* that control a large transcriptional network of 162 genes involved in the cellulose, glucomannan, glucan, xylan, and pectin degradation and downstream utilization of the released oligo- and monosaccharides (Fig. 1). Below, the reconstructed regulons and catabolic pathways are described in more detail.

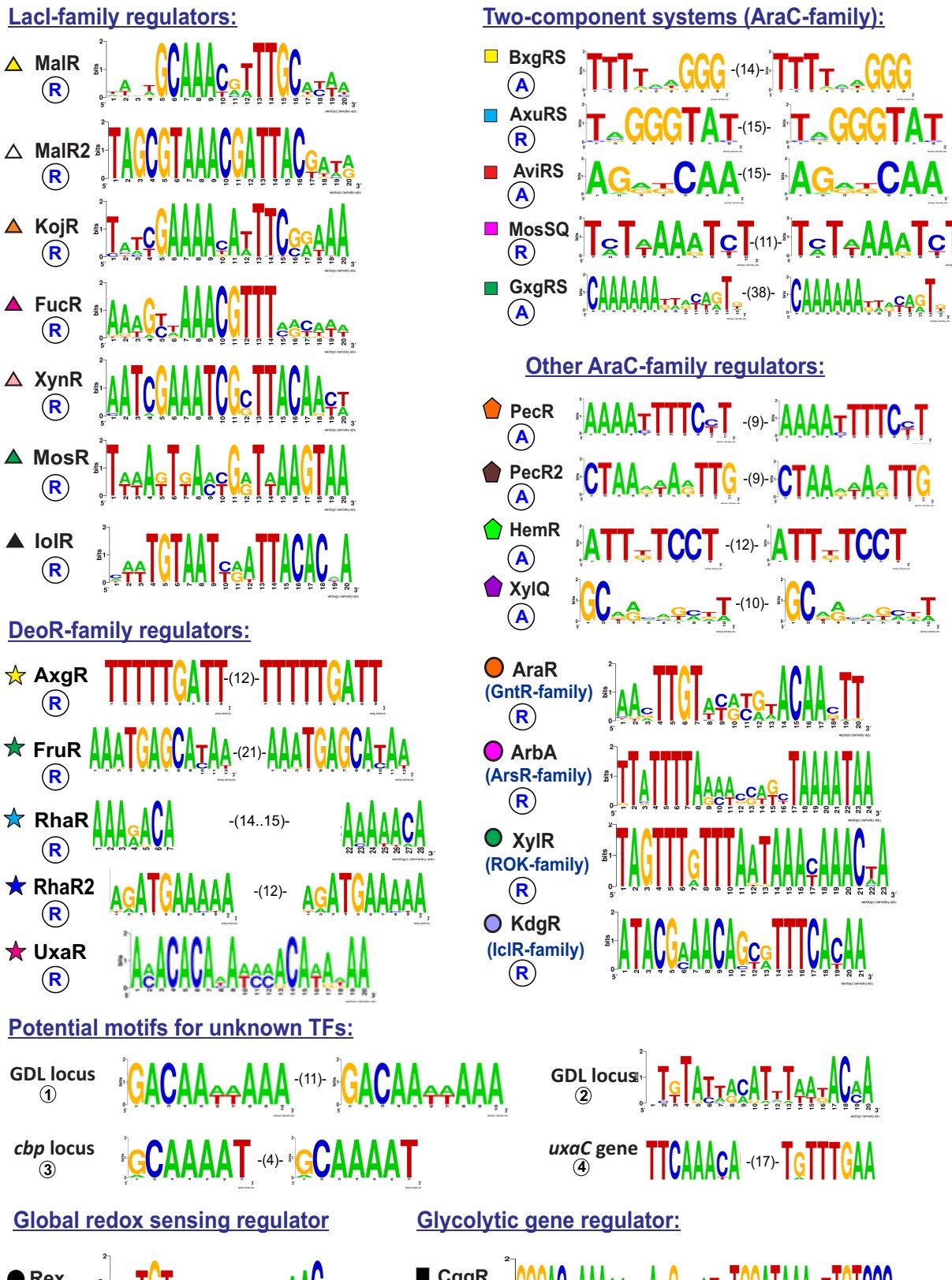

**FIG 2** DNA-binding motifs for reconstructed carbohydrate utilization regulons. TFBS sequence logos were built by WebLogo using all candidate TFBSs identified by comparative genomics techniques in *C. bescii* and related *Caldicellulosiruptor* genomes. The identified TFBSs

**(iv) Cellulose and mannan degradation.** The glucan degradation locus (GDL) is essential for plant biomass degradation and encodes six multidomain GHs/CBMs that catalyze the initial breakdown of $\beta$-glucans (such as cellulose) and mannans to generate an extracellular pool of cellulose oligosaccharides (COS) and mannose oligosaccharides (MOS), respectively (Fig. 1). These oligosaccharides are presumably delivered to the cytoplasm by CUT1-family ABC transporters, AviABCD and MosABC, predicted to be activated by local TCRS regulators AviRS and MosSQ, respectively (Fig. 3). The *avi* gene cluster contains an additional AviRS-regulated gene, *celD*, encoding a secreted bifunctional endoglucanase/endoxylanase. Cytoplasmic $\beta$-glucosidase BglA and two cellobiose phosphorylases (Cbp) required for COS utilization are encoded in a gene locus preceded by a new direct repeat DNA motif for an unknown TF (see Fig. S1). The mannan-derived MOS residues are degraded by cytoplasmic mannosidases (MnnA and MnnB) and phosphorylases (MosP and ManP). The novel LacI-family regulator MosR coregulates the MosABC transporter with the MOS phosphorylase MosP and cellobiose 2-epimerase MosE (Fig. 3), to convert 4-*O*-$\beta$-D-glucopyranosyl-D-mannose (epicellobiose) to cellobiose (34). As such, the *mos* gene cluster is involved in the utilization of oligosaccharides derived from degradation of mannan and glucomannan.

**(v) Pectin degradation.** The last three genes in the GDL encode three extracellular pectate lyases—PecA, PecB, and PecC—that are involved in pectin degradation and are presumably coregulated by a novel regulator from the AraC family, which is encoded by the upstream gene *pecR* (Fig. 3). Degradation of polygalacturonan and rhamnogalacturonan, two major pectin polymers, results in the release of pectin oligosaccharides (POS) and rhamnogalacturonides, respectively. The derived oligosaccharides are taken up via three predicted CUT1-family ABC transporters and further metabolized using an array of cytoplasmic CAZymes and catabolic enzymes (Fig. 1) that are encoded by several gene clusters and presumably controlled by at least six local TFs (Fig. 3).

Unsaturated galacturonate is released from rhamnogalacturonate by RhiN and utilized via a 2-keto-3-deoxy-D-gluconate (KDG) intermediate via a dedicated pathway controlled by the KDG repressor KdgR from the IclR family. Saturated galacturonate is converted to tagaturonate via the isomerase UxaC; however, other enzymatic steps from conventional galacturonate catabolic pathways are missing in *Caldicellulosiruptor* spp. (24). The predicted regulator UxaR (DeoR family) coregulates two uncharacterized sugar catabolic gene clusters, *hem* and *uxa* (Fig. 3), encoding putative tagaturonate utilization enzymes (Fig. 1). Two predicted POS transporter operons, *pecXYZ* and *hemXYZ*, are likely controlled by local regulators from the AraC family, PecR2 and HemR, respectively. The predicted rhamnogalacturonide transporter RglFGH is coexpressed with secreted rhamnogalacturonan exolyase RglX. However, their genomic locus lacks a dedicated TF. Expression of the rhamnose catabolic genes *rhaA*, *rhaB*, and *rhaD1* is possibly controlled by two DeoR-family paralogs, RhaR and RhaR2. L-Lactaldehyde, a toxic intermediate of rhamnose catabolism, is further metabolized to L-lactate by aldehyde dehydrogenase or to 1,2-propanediol by lactaldehyde reductase (30), although both enzymes are missing in *Caldicellulosiruptor* genomes. This final step of rhamnose catabolism is likely catalyzed by a novel three-subunit dehydrogenase, encoded in Athe_0702-4 from the RhaR2-regulated gene cluster, which is homologous to acetoin dehydrogenase.

**(vi) Xylan degradation.** Xylose is a major component of $\beta$-xylans, while arabinoxylan (AX) and glucuronoxylan (GX) contain the side chains of arabinose and glucuronic acid, respectively. In the *C. bescii* genome, 10 multigene loci and four single gene loci encode CAZymes, transporters and regulators involved in xylan degradation, uptake and utilization of resulting oligosaccharides, and catabolism of xylose, arabinose, and glucuronate (Fig. 4). The ROK-family regulator XylR from *C. bescii* is 33% identical

**FIG 2** Legend (Continued)
have either inverted-repeat (palindromes, represented as a single DNA element) or direct-repeat (tandems, represented by two identical DNA elements with a nonconserved linker of specified length) structures. Predicted repression or activation mechanism of regulation for each TF is specified by circled "R" and "A" letters, respectively.

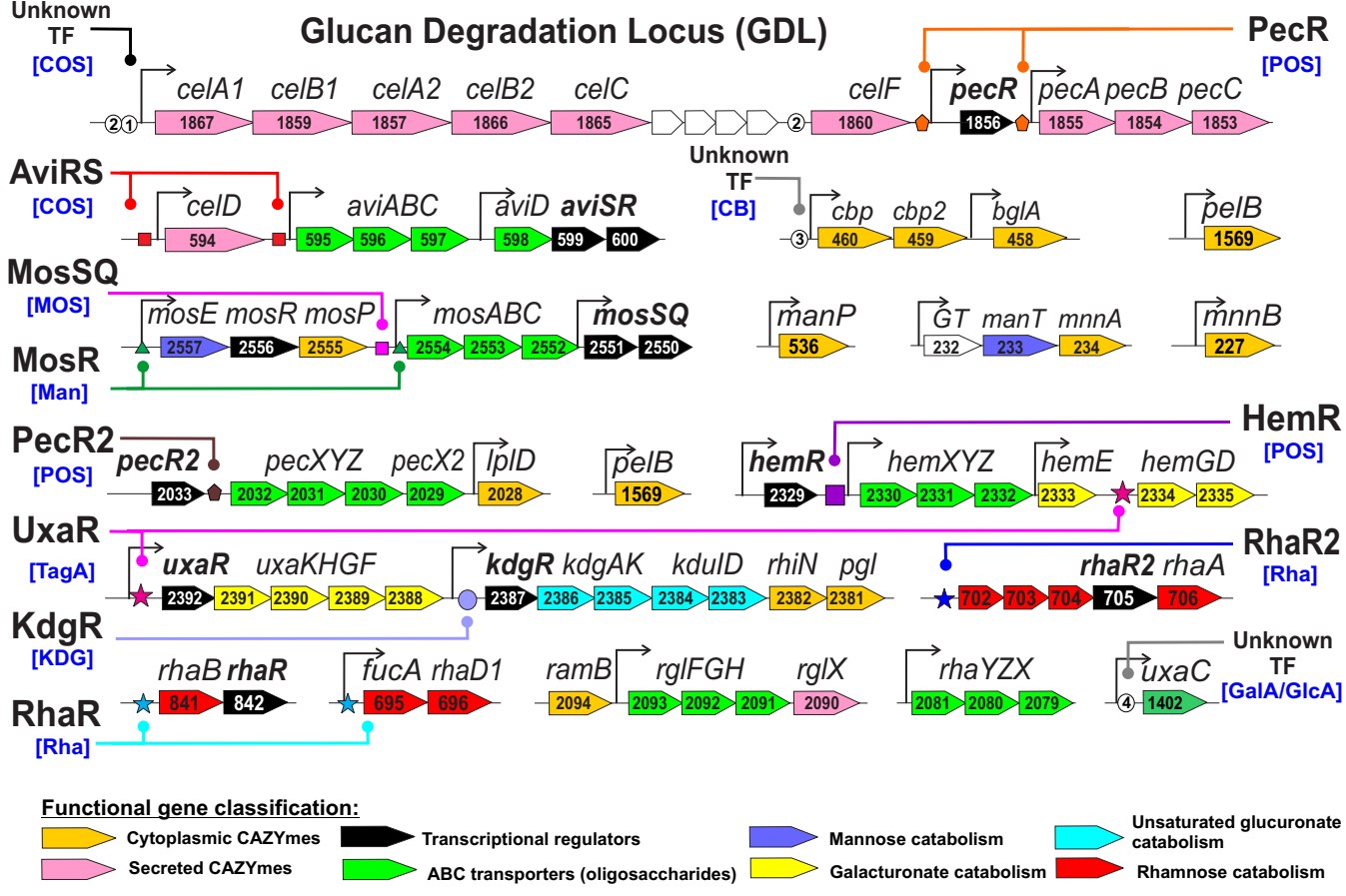

**FIG 3** Genomic context of reconstructed regulons for catabolic and transporter genes involved in glucan, cellulose, mannan and pectin degradation in *C. bescii*. Carbohydrate utilization genes are indicated by arrows with colors according to the functional gene classification and numbers corresponding to their locus tag with Athe_ prefix. Candidate TF binding sites are indicated by colored symbols and connected to a cognate TF. Putative DNA-binding sites of as-yet-unknown TFs are indicated by circled numbers. Predicted effectors of TFs are given in square brackets. Transcription start sites (TSSs) identified by in RNA-Seq experiments are indicated by standing arrows. Genes upregulated on a particular substrate according to RNA-Seq and their corresponding log₂-fold change are shown in Fig. S3.

(amino acid level) to the previously characterized xylose-responsive repressor from *T. maritima*, which coregulates xylan degradation, XOS and xylose utilization genes (22). In contrast to *T. maritima*, the reconstructed XylR regulon in *C. bescii* contains only two target loci – the xylose isomerase *xylA* and the XOS utilization gene cluster *xynUVW-xylR-xynA*. The xylan degradation locus (XDL) (Athe-0174-Athe_0185), which encodes two CUT1-family ABC transporters (AxoFGE and XloEFG) and seven CAZymes (three secreted and four cytoplasmic), is presumably controlled by two TFs, a novel LacI-family regulator, named XynR, and the arabinose repressor, AraR. The remaining xylan and XOS catabolic gene loci, the xylose catabolism operon *xynR-xylB*, the glucuronidase *aguX*, and glucuronate catabolic genes *uxuAB*, are coregulated via the reconstructed XynR regulon (Fig. 4). The latter glucuronate utilization genes belong to a larger GX utilization gene cluster, which also encodes a CUT1-family ABC transporter (BxgLFG) and a TCRS regulator (BxgRS), that are presumably involved in utilization of glucuronoxylan oligosaccharides (GXOS) and locally controlled by BxgRS characterized by two conserved TFBSs with tandem repeat structure (see Fig. S1).

The α-xyloside utilization gene locus, containing a CUT1-family ABC transporter (AxuABC) and cytoplasmic α-xylosidase XylS, encodes two novel TFs from the AraC-family (named AxuRS and XylQ), that are predicted to control *axuABC* and *xylSQ* genes, respectively (Fig. 4). Another gene cluster, containing a cytoplasmic endo-α-1,5-L-arabinanase (AbnA), a CUT1-family ABC transporter (AxeUVW), and a TCRS regulator (AxeRS), is likely involved in utilization of arabinoxylo-oligosaccharides (AXOS).

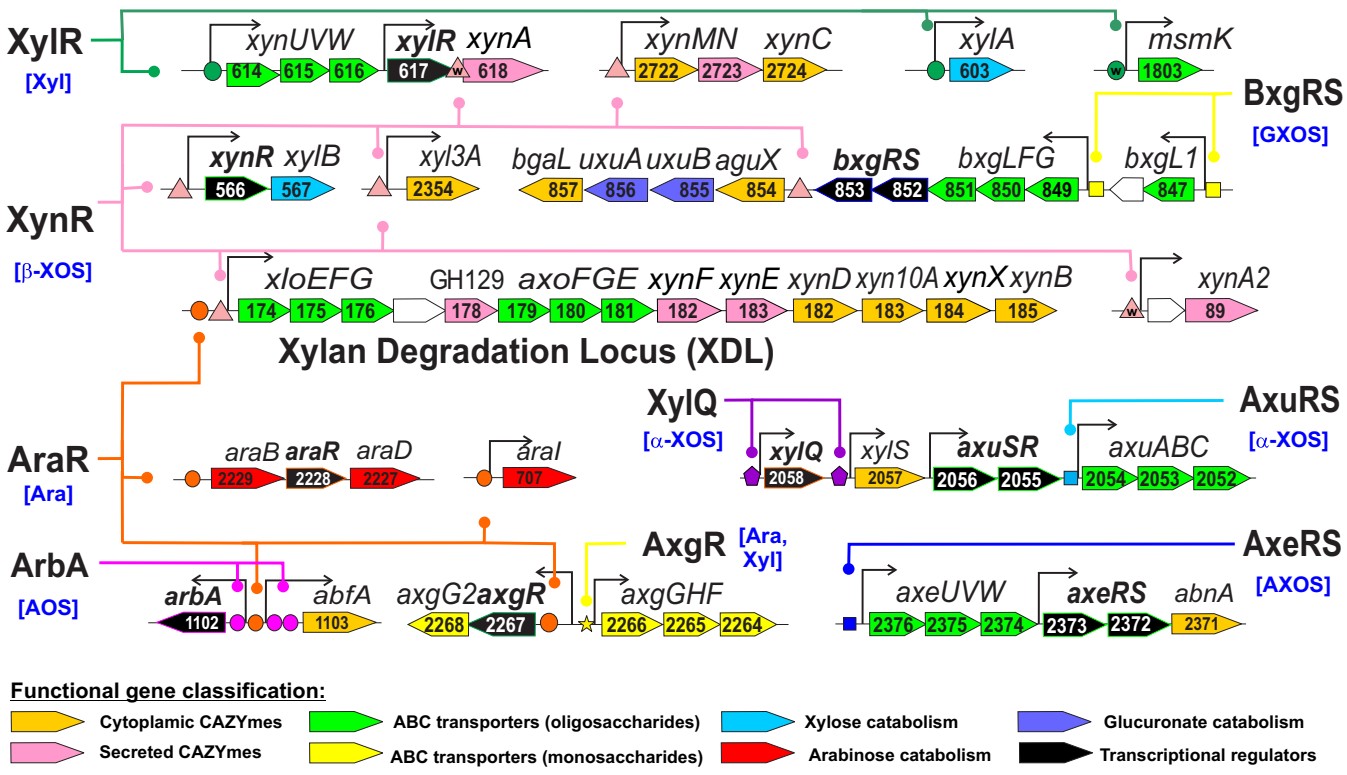

**FIG 4** Genomic context of reconstructed regulons for catabolic and transporter genes involved in xylan degradation in *C. bescii*. For abbreviations, see the Fig. 3 legend.

The AraR regulator from the GntR family is 42% identical (amino acid level) to arabinose-responsive repressors from *B. subtilis* and *Clostridium acetobutylicum* (35, 36). The reconstructed AraR regulon in *C. bescii* includes the arabinose catabolic operon *araBRD* and the novel predicted L-arabinose isomerase *araI*, the above-mentioned xylan degradation gene locus, the α-L-arabinofuranosidase *abfA*, and the *axg* operon encoding a CUT2-family ABC transporter for monosaccharides. The *abfA* gene in *C. bescii* and other *Caldicellulosiruptor* spp. is divergently transcribed with a hypothetical gene encoding novel ArsR-family regulator (ArbA) (Fig. 4). A common intergenic region of *abfA* and *arbA* contains a candidate AraR-binding site and three predicted ArbA-binding sites that are conserved across the genus (see Fig. S1). The AraR-controlled *axg* locus also encodes a candidate local regulator of the *axgGHF* operon (AxgR) that belongs to the DeoR family of TFs. The *axgR-axgGHF* intergenic region contains a putative AxgR regulatory site with tandem repeat structure (Fig. 2), which is conserved across all eight *Caldicellulosiruptor* genomes possessing the *axg* loci (see Fig. S1). Thus, both AraR and XynR TFs function as master regulators controlling five and six gene loci in *C. bescii*, respectively, while seven other identified TFs associated with the xylan degradation and utilization loci are presumably local.

**(vii) α-Glucan degradation.** Linear and branching alpha-glucans, containing D-glucose monomer linked together through either α-1,4 or α-1,6 glycosidic linkages (i.e., amylose and amylopectin), are two major constituents of starch. *Caldicellulosiruptor* genomes contain a conserved set of extracellular hydrolases involved in the primary degradation of α-glucans (AmyA2, PulA, and AmyX), the CUT1-family ABC transporter MalEFG for uptake of maltooligosaccharides (MOS), and cytoplasmic CAZymes for MOS degradation (NplT, Cga, AmyA, MalL, and PulA2), resulting in maltose and glucose as the final products (Fig. 1). Maltose is further utilized via the GlgP phosphorylase. The LacI-family regulator MalR controls most of the genes and operons involved in α-glucan degradation (*pulA-amyX*), MOS uptake and catabolism (*malFG-glgP-malR*, *malE*, *amyA*, *nplT*, and *cga*). All of these genes are coregulated by a common palindromic

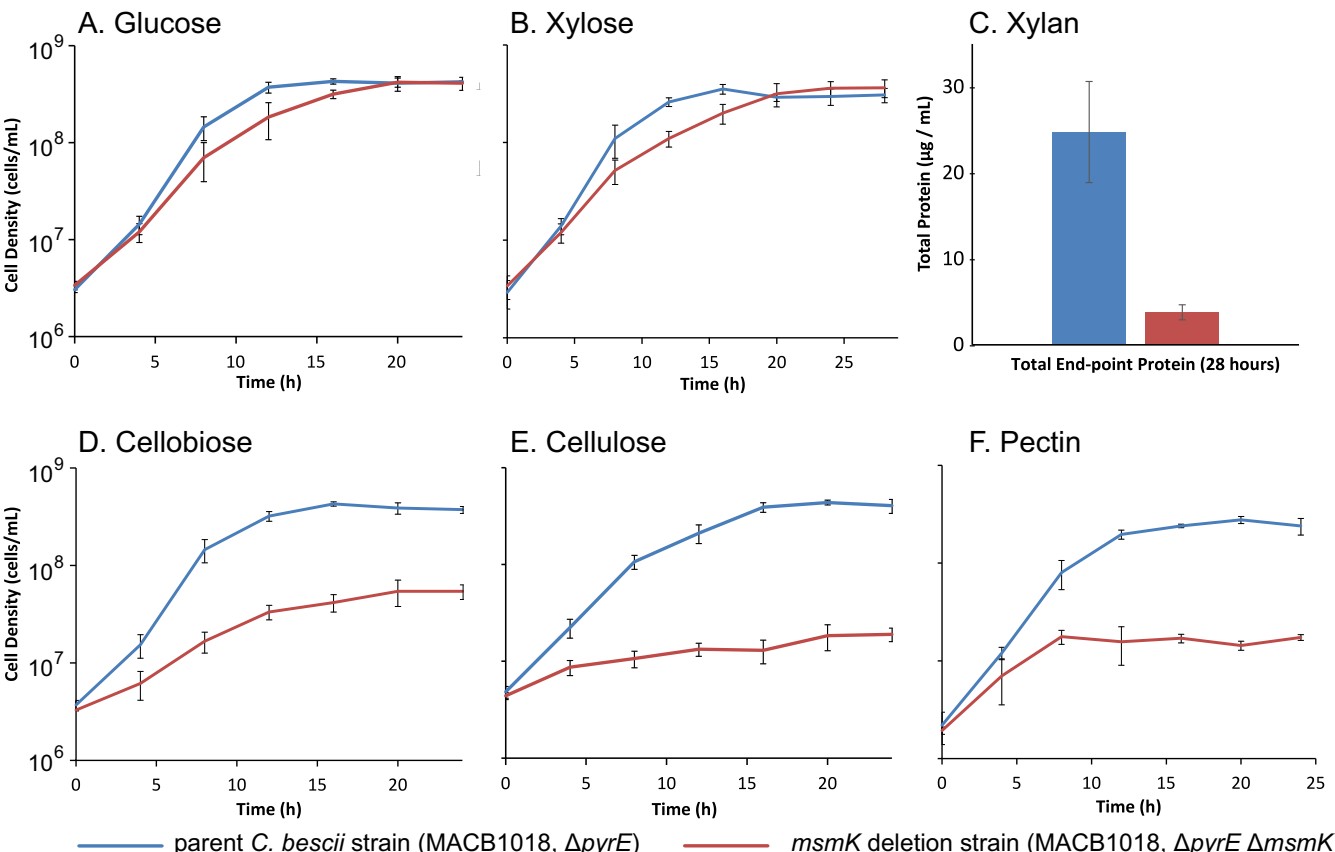

**FIG 5** Comparison of growth for the Δ*msmK* strain and the parent strain on various carbon substrates. The Δ*msmK* strain and the parent strain were grown in biological triplicate (*n* = 3) at 75°C on a variety of soluble and insoluble carbohydrate substrates. Growth was compared on glucose (A), xylose (B), xylan (C), cellobiose (D), cellulose (E), and pectin (F). Insoluble xylan particles with similar scale to *C. bescii* cells interfered with direct quantification of growth (counting cells). Therefore, for growth on xylan, the growth was reported as the endpoint (28 h) protein concentration. For all other substrates, growth was reported as the cell density (cells/ml) over the course of growth (24 to 28 h).

DNA motif (Fig. 2; see also Fig. S1), which is similar to binding motifs of orthologous MalR repressors from *Bacillus* (37), *Streptococcus* and *Lactobacillus* (38), *Bacteroides* (17), and *Bifidobacteria* (16). An additional LacI-family regulator (KojR) presumably controls the Athe_0397-0403 gene cluster, containing cytoplasmic phosphorylases (*pgmB-kojP* and *treP*) and the CUT1-family ABC transporter (*kojEFG*) involved in the utilization of α-glucoside disaccharides trehalose and kojibiose.

**Characterization of ATPase component for oligosaccharide transporters.** As shown above, *Caldicellulosiruptor* spp. possess a large number of candidate oligosaccharide transporters from the CUT1 family of ABC transporters (Fig. 1). These include 12 systems implicated in the uptake of oligosaccharides derived from the degradation of cellulose, glucomannan, pectin, and xylans, as well as transporters for maltose/MOS, the disaccharides trehalose and kojibiose, and deoxynucleosides (Table 1). All these CUT1-family transporters lack a dedicated ATP-binding ATPase component and presumably use a shared ATPase encoded by the *msmK* gene (Athe_1803). To test this hypothesis, we constructed the *msmK* deletion mutant strain of *C. bescii* (see Fig. S2) and characterized its growth on mono- and polysaccharides.

Under standard growth conditions on the monosaccharide substrates xylose and glucose, the Δ*msmK* strain grew to the same final density as the parent strain (Fig. 5). On both substrates, while the same final cell density was reached, a minor growth phenotype was observed whereby the growth rate was lower during log phase than the parent strain. During growth of the parent strain on both glucose and xylose, maximum cell density was reached at 16 h, while the deletion strain did not reach maximum density until 20 h. Growth of the Δ*msmK* strain was severely impaired relative to

the parent strain on all tested di- and oligosaccharide substrates (Fig. 5). Substrates that supported diminished growth capacity were cellobiose, crystalline cellulose, xylan, and pectin. The growth phenotype was the least severe on cellobiose, where the final cell density was roughly 8-fold lower than the parent strain. During growth on cellulose and pectin, the final cell densities were over an order of magnitude lower than the parent strain. These growth phenotypes of the Δ*msmK* strain of *C. bescii* suggest that MsmK is a functionally promiscuous ATPase, which is essential for utilization of di- and oligosaccharides derived from degradation of cellulose, pectin and xylan; however, MsmK is not required for monosaccharide uptake.

**Reconstruction of central carbohydrate metabolism and regulons.** The reconstructed carbohydrate catabolic pathways produce phosphorylated hexoses, pentoses, or trioses that are further catabolized through the central carbohydrate metabolism pathways (Fig. 1). A total of 18 genes encoding 16 enzymes from glycolytic and the pentose phosphate pathways are conserved in all analyzed *Caldicellulosiruptor* spp. (see Data Set S2). The central glycolytic gene repressor CggR controls the *cggR-gap-(pgk-tpi)-pgm-eno* operon encoding enzymes from the central part of glycolysis. In *Bacillus subtilis*, the CggR repressor binding to its DNA operator is modulated by fructose-1,6-bisphosphate (39). By analyzing the upstream gene regions of *cggR* genes, the putative CggR-binding motif, an inverted repeat with sequence GGGAC-(27)-GTCCC (see Fig. S1), can be identified that is similar to DNA motifs of CggR regulators from other *Firmicutes*, e.g., *Bacillus* spp. (37). In addition to the conventional glyceraldehyde-3-phosphate (GAP) dehydrogenase (encoded by the CggR-controlled *gap* gene), all known *Caldicellulosiruptor* species have an alternative ferredoxin-dependent GAP oxidoreductase (GOR) (40). GOR in *C. bescii* is encoded by the two-gene *gorSL* operon and controlled by a global redox-sensing regulator Rex that responds to the intracellular ratio of $NADH:NAD^+$ (13). Among other experimentally determined Rex regulon members in *C. bescii* are pyruvate ferredoxin oxidoreductase (POR), ferredoxin-dependent membrane-bound [NiFe]-hydrogenase (MBH), and the bifurcating [FeFe]-hydrogenase (BF-$H_2$ase).

Comparative genomics was used to reconstruct the Rex regulon in the genomes of *Caldicellulosiruptor* species. The identified 20-bp palindromic motif was similar to previously identified TFBS motifs of Rex in other clostridial lineages (41). A positional-weight matrix was constructed to scan the 14 *Caldicellulosiruptor* genomes for additional candidate Rex-binding sites. After the removal of false-positive sites by a consistency check approach, the predicted members of Rex regulon in *C. bescii* and related species were identified. The most conserved members of the reconstructed Rex regulon were involved in central carbohydrate metabolism (GOR, POR, BF-$H_2$ase, and MBH), acetate fermentation (Pta and AckA) and serine metabolism (SerC and SerA) (Fig. 1; see also Data Set S4). Candidate Rex-binding sites upstream of these operons were conserved in all analyzed *Caldicellulosiruptor* genomes (see Fig. S1). Among additional Rex regulon members in *C. bescii* that were also conserved in most other species, a second copy of putative POR-encoding operon was found, an hypothetical operon encoding sulfur carrier protein TusA, CoA-disulfide reductase, an Fe-S oxidoreductase and phosphoglucosamine mutase.

**Carbohydrate-specific transcriptomic responses of *C. bescii*.** To assess the transcriptional responses of *C. bescii* to glucose, cellobiose, cellulose, xylose, and xylan, transcriptome sequencing (RNA-Seq) analyses were done for mid to late exponentially growing cultures grown in triplicates on each of five carbon sources. Reads per samples ranged from 3.6 to 5.0 million with an average 4.2 million reads across the 15 samples (see Data Set S3A and B). Averaged normalized expression values on $\log_2$ scale (see Data Set S3C) were compared between each pair of conditions to estimate number of differentially transcribed genes in these experiments (Table 2). Overall, 277 *C. bescii* genes were upregulated at least 4-fold ($\log_2$-fold $> 2$) for at least one pair of conditions. Growth on polysaccharides cellulose and xylan promoted larger transcriptional responses (64 to 86 upregulated genes) compared to glucose, xylose, and cellobiose (10 to 25 upregulated genes). These upregulated genes were further classified into

**TABLE 2** Numbers of differentially transcribed genes for *C. bescii* growth on various carbohydrate substrates

| Substrate | No. of differentially transcribed genes[a] | | | | |
| | Cellulose | Cellobiose | Glucose | Xylan | Xylose |
|---|---|---|---|---|---|
| Cellulose (Avi) | | 86 (49) | 68 (35) | 60 (17) | 81 (31) |
| Cellobiose (CB) | 10 (0) | | 18 (12) | 23 (5) | 31 (7) |
| Glucose (Glc) | 25 (4) | 23 (21) | | 30 (13) | 43 (8) |
| Xylan (XN) | 37 (15) | 70 (50) | 64 (42) | | 65 (39) |
| Xylose (Xyl) | 14 (2) | 23 (11) | 22 (9) | 10 (5) | |

[a]The numbers represent differentially transcribed genes with a $\log_2$-fold change of $>2.0$ for comparisons between column and corresponding line. The numbers in parentheses correspond to subsets of differentially expressed genes that are involved in the reconstructed carbohydrate utilization pathways/regulons. For example, 86 genes were upregulated on cellulose compared to cellobiose; of these, 49 genes belong to the reconstructed pathways/regulons. The functional classifications of the remaining non-CU genes are provided in Data Set S3D.

three groups (see Data Set S3D): (i) 98 genes (36 operons) that belong to the reconstructed CU pathways and TF regulons; (ii) 93 genes involved in other metabolic pathways, including amino acid and nucleotide metabolism (25 genes), peptidases (4 genes), and ABC efflux transporters (9 genes); and (iii) 87 hypothetical genes, including pseudogenes and transposases. The transcriptional responses of CU genes involved xylan, glucan, cellulose, mannan, and pectin degradation genes were placed into the context of the reconstructed TF regulons (Fig. 1 and Table 3; see also Fig. S3).

**(i) Cellulose and mannan degradation.** Extracellular GHs encoded by the GDL locus are required for degradation of both $\beta$-glucans (e.g., cellulose) and mannans. The GDL genes were upregulated (4- to 7-fold) on cellulose and xylan (see Fig. S3). Despite the fact that this locus lacks a dedicated TF, the identified common DNA motifs in the *celA1* and *celF* promoter regions (Fig. 2) suggest potential involvement of as-yet-unknown TFs to the observed transcriptional responses to both polysaccharides. The

**TABLE 3** Differential expression of carbohydrate utilization genes in *C. bescii* grown on carbohydrate substrates

| Target operon(s)[a] | DE substrate(s)[b] | CU pathway(s)[c] | TF regulon(s)[d] |
|---|---|---|---|
| *xynUVW*, *xylA* | XN, Avi, Xyl | Xylan | XylR |
| *xyl3A*, *xynM-xynN-xynC*, *xynA*, *X-xynA2* | XN, Avi | Xylan | XynR |
| *xynR-xylB* | XN | Xylan | XynR |
| *xloEFG-X-GH129-axoFGE-xynF-xynE-xynD-xyn10A-xynX-xynB* | XN, Avi | Xylan (XDL) | XynR, AraR |
| *aguX-uxuB-uxuA-bgaL* | XN, Avi | Xylan, glucuronate | XynR |
| *xylR* | XN | Xylan | |
| *msmK* | XN, Avi, CB | Oligosaccharides | XylR |
| *bxgL-bxgFG* | XN, Avi | Xylan | BxgRS |
| *bxgL1-X* | XN, Avi, Xyl, Glc | Xylan | BxgRS |
| *axuABC* | XN, Avi | Xylan, $\alpha$-xylosides | AxuRS |
| *celA1-celB1-X-celA2-celB2-celC*; *celF* | Avi, XN | Cellulose (GDL) | New motifs 1/2 |
| *celD*; *aviABC* | Avi, CB, XN, Xyl | Cellulose | AviRS |
| *cbp-cbp2* | Avi, CB, XN | Cellobiose | New motif 3 |
| *mosABC* | Avi, CB, XN | Mannose OS | MosR, MosSQ |
| *axgGHF* | XN, Avi, Xyl, Glc | Monosaccharides | AxgR, AraR |
| *gxgABC* | Glc, XN, Xyl | Monosaccharides | GxgRS |
| *pulA-amyX* | Avi, CB, Glc | Starch | MalR |
| *malE*; *malFG-glgP-malR*; *amyA*; *cga* | Avi, CB, Glc | Maltodextrin OS | MalR |
| *hemXYZ* | XN | Pectin OS | HemR |

[a]CU operons with similar transcriptional response, assigned catabolic pathways, and TF regulons are grouped together and separated by semicolon. Detailed information on CU operon-containing genes including their locus tags and functional annotations is provided in Data Set S1. Additional differentially expressed genes involved in the utilization of mannose OS, pectin, fructose, inositol, and deoxynucleosides are listed in Data Set S3D and visualized in Fig. 1.
[b]Five analyzed substrates: XN, xylan; Avi, crystalline cellulose; CB, cellobiose; Xyl, D-xylose; Glc, D-glucose. Substrates that demonstrated differential expression (DE) of genes from target operons with $\log_2$-fold change of $>2.0$ are shown. Substrates and genes with moderate differential expression that showed $\log_2$-fold change between 1.5 and 2 are underlined. Detailed information on differential gene expression values is provided in Data Set S3.
[c]Metabolic scheme of the reconstructed CU pathways is given in Fig. 1.
[d]TF regulons with predicted TFBSs located in the promoter regions of target operons are indicated. Putative TFBS motifs for yet unknown TFs are numbered as 1, 2, and 3. Consensus sequences for identified TFBS motifs are given in Fig. 2.

*aviABC* and *celD* genes from the COS utilization locus were upregulated on cellulose, cellobiose, and xylan (10- to 15-fold) and presumably controlled by their cognate TCRS regulator AviRS. The *cbp-cbp2* operon encoding two cellobiose phosphorylase paralogs was ~6-fold upregulated on cellulose, cellobiose, and xylan compared to glucose-grown cells and was associated with a novel direct repeat DNA motif for an as-yet-unknown TF (see Fig. S1). The *mosABC* operon was highly upregulated on cellulose and cellobiose (50- to 90-fold) and also moderately responded to xylan (9-fold), and the cognate TCRS MosSQ is presumably involved in the induction of this mannose OS transporter.

**(ii) Xylan degradation.** Transcription levels of all 25 genes from the six operons that constitute the reconstructed XynR regulon were significantly elevated in *C. bescii* grown on xylan compared to cells grown on cellobiose. The highest fold changes of gene expression (~10- to 100-fold) were observed for the xylan degradation genes *xynA2*, *xyl3A*, and *xynMNC* and the proximal XDL genes, while the distal XDL genes, as well as *xynR-xynB* and *aguX-uxuBA-bgaL*, were moderately upregulated on xylan (4- to 9-fold). In addition, most of the XynR-regulated genes demonstrated modest upregulation on cellulose. The *xylUVW* and *xylA* genes from the XylR regulon were highly upregulated on xylan, xylose, and cellulose. A low-scoring XylR-binding site was identified upstream of *msmK* encoding an ATPase component of CUT1-family transporters (see Fig. S1). *msmK* is highly expressed in all tested conditions, with only 3- to 4-fold higher expression in *C. bescii* grown on xylan and cellulose compared to glucose-grown cells (see Fig. S3), suggesting the XylR repressor has a weak effect on its expression. Also highly upregulated on xylan were *bxg* and *axu* encoding putative GX oligosaccharides and α-xyloside transporters that are presumably regulated by their local TCRS, BxgRS and AxuRS, respectively.

**(iii) Monosaccharide transporters.** *C. bescii* grows on various monosaccharides, including fructose, glucose, galactose, mannose, xylose, arabinose, fucose, and rhamnose (42, 43). Fructose uptake and utilization are mediated via the dedicated phosphotransferase system (PTS) FruAB-PtsI-Hpr and phosphofructokinase FruK that are controlled by the DeoR-family repressor FruR. Two candidate ABC transporters from the CUT2 family, AxgFGH and GxgABC, are potentially involved in uptake of other monosaccharides in *C. bescii* and are presumably controlled by their local transcription factors, the AraC-family regulator AxgR and the TCRS regulator GxgRS, respectively. Each of these systems includes a dedicated ATPase (GxgA and AxgG), which agrees with experiments showing that the MsmK ATPase is not essential for monosaccharide uptake. The proposed sugar specificities of these two monosaccharides transporters can be predicted using transcriptomics data. In *C. bescii*, both *axgFGH* and *gxgABC* genes are upregulated on glucose (10- to 25-fold) and xylan (5- to 10-fold), although their basal levels of expression and responses to other carbohydrates were different (see Data Set S3). While the *axgFGH* genes were highly upregulated on xylose (30- to 100-fold) and cellulose (15- to 30-fold), the *gxgABC* operon was highly expressed on all carbon sources, with only 3-fold induction on xylose compared to cellobiose-grown cells (minimal expression level). In *C. saccharolyticus* (10), an ortholog of the *axgFGH* transporter was induced on arabinose, xylose, and galactose, while the *gxgABC* ortholog was upregulated on galactose, xylose, and glucose (see Data Set S1B). These results suggest that the cognate AxgR and GxgRS regulators for these monosaccharide transporter operons respond to different carbohydrates.

**(iv) Starch and maltodextrin degradation.** The starch degradation operon *pulA-amyX*, the maltodextrin OS transporter *malEFG* and cytoplasmic enzymes *amyA* and *cga* were moderately upregulated on cellulose, cellobiose and glucose compared to xylan and xylose grown cells (3- to 10-fold). MalR is a predicted global regulator of all these starch/maltodextrin utilization genes that can be partially responsible for their similar transcriptional responses.

**5′-Transcription start site mapping for CU genes.** We used 5′-end RNA sequencing for all five *C. bescii* cultures from RNA-Seq experiments to globally determine the 5′ transcription start sites (5′TSSs) at nucleotide resolution. As result, 5′TSSs were

mapped within the upstream regions of 83 of 100 putative operons containing CU genes, as well as for 18 putative operons encoding enzymes from the central carbohydrate metabolism, including all predicted Rex- and CggR-regulated operons (see Data Set S2). We further performed multiple alignments of the 5′TSS-containing gene upstream regions from *C. bescii* and related *Caldicellulosiruptor* species and searched for associated candidate −35 and −10 promoter elements (see Fig. S1). The majority of identified 5′TSSs were located within 180 bp upstream of the translation start site (median distance 50 bp). However, the *fucI-alfA* operon has the largest 5′-untranslated region (5′ UTR) between the transcription and translation start sites (329 bp), containing tandem FucR-binding sites, suggesting FucR acts as a repressor.

To determine whether TFs in CU from the reconstructed regulons were activators or repressors, the positions of their predicted TFBSs were compared to the experimentally mapped 5′TSSs in *C. bescii* (see Fig. S1). All 12 TFs from the LacI and DeoR families, as well as XylR (ROK family), ArbR (ArsR family), AraR (GntR family), KdgR (IclR family), and CggR (SorC family), have their cognate TFBSs located within 5′ UTRs or overlapping candidate promoter elements, suggesting that they are repressors (Fig. 2). This is in agreement with the previously reported repression mechanism for other TFs from the protein families discussed above (16, 18, 22, 30, 32, 33, 44). In contrast, all four single-component regulators, as well as three of five TCRS containing DNA-binding domains from the AraC family, have their cognate DNA sites (tandem repeats with large spacers) located upstream of candidate 5′TSS-associated promoter elements, suggesting that they are activators. In fact, most characterized regulators in this family (44), including a novel TCRS for xyloglucan degradation in *Ruminiclostridium cellulolyticum* (45), function as transcriptional activators.

Many global regulators of carbohydrate metabolism in bacteria, including Rex, are capable of acting as both repressors or activators, depending on the location of their binding sites with respect to the promoter elements of their target genes (27, 41, 46). For the majority of core conserved Rex regulon members, Rex-binding sites either overlap with—or are located downstream of—the promoter elements associated with 5′TSSs, suggesting that Rex functions as a transcriptional repressor of these operons (see Fig. S1). However, the MBH-encoding operon has a Rex operator located 45 bp upstream of its 5′TSSs, suggesting a potential activation mechanism. The hypothesis that Rex is an activator of MBH is supported by decreased MBH transcript expression, as previously determined in an RNA-Seq analysis of a *rex* deletion strain of *C. bescii* (13). Thus, these results suggest that the NADH-responsive Rex regulator controls expression of two hydrogenases in an opposite manner: namely, Rex represses the NADH-dependent cytoplasmic BF-H$_2$ase and activates the membrane-bound and NADH-independent MBH, suggesting an adaptation mechanism to fine-tune the NADH concentration in the cell.

**Conclusions.** Extremely thermophilic cellulolytic bacteria from the genus *Caldicellulosiruptor* possess extensive and highly diversified CU machinery. *C. bescii* is a promising nonmodel species for development of a microbial metabolic engineering platform for conversion of plant biomass to next-generation bioproducts. Accurate functional assignment of CU genes across diverse bacteria with complete genomes is challenging due to frequent variations of the respective pathways. Here, a subsystems-based comparative genomic approach was used in conjunction with available experimental data on genome-wide gene expression to reconstruct CU pathways and transcriptional regulons in *C. bescii*. The global CU model so developed includes >150 protein families (metabolic enzymes, transporters and transcriptional regulators) spanning >20 distinct pathways. Using a comparative analysis of 14 *Caldicellulosiruptor* genomes, DNA-binding motifs and reconstructed CU regulatory networks were identified, including both previously known regulons that have orthologous TFs characterized in other *Firmicutes* (e.g., AraR, KdgR, and XylR), as well as *ab initio* predicted novel TF regulons from LacI, DeoR, AraC, and other families. Reconstructed regulons provided an additional layer of genome

mSystems®

context, helping to significantly improve the accuracy of functional annotations and metabolic reconstruction. For instance, the inferred AraR regulon includes a novel functional variant of an arabinose isomerase gene, which is nonorthologous to previously characterized enzymes. The reconstructed regulatory network was validated by gene expression profiling via RNA-Seq analysis of *C. bescii* grown on glucose, cellobiose, cellulose, xylose, and xylan. In addition, the TSS data for *C. bescii* were used to delineate the transcriptional architecture of CU genes and to propose the activation or repression mechanism of regulation for predicted TF regulons. Results of previous gene expression experiments in *C. bescii* and the related *C. saccharolyticus* species grown on various mono- and polysaccharides were also considered. The observed upregulation of genes involved in catabolism of specific carbohydrates showed excellent correlation with the reconstructed content of transcriptional regulons.

Among other insights, it was determined that, while the CUT2 family ABC transporters (putatively responsible for monosaccharide transport) are each chromosomally associated with a dedicated ATPase subunit, the CUT1 family transporters each lack an associated ATPase. A similar scenario has been previously observed in *Streptococcus pneumoniae*, where *msmK* was characterized as a shared ATPase for six different CUT1 family ATP transporters that each lacked their own ATPase (47). Likewise, in *B. subtilis*, a homolog (*msmX*) was demonstrated to be a multipurpose ATPase involved in the uptake of pectin oligosaccharides (48). Deletion of *msmK* in *C. bescii* resulted in the predicted phenotypes on all tested carbohydrate substrates, indicating that this ATPase is indeed functionally promiscuous.

This study elucidated the transcriptional regulatory network and mechanisms controlling expression of CU genes in *C. bescii*, thereby improving the functional annotations of carbohydrate transporters and catabolic enzymes to inform a metabolic model of *C. bescii* (31). The developed *C. bescii* model containing the targeted reconstruction of CU pathways supports metabolic engineering strategies for this biotechnology-important thermophilic bacterium capable of unpretreated lignocellulose conversion to bioproducts.

## MATERIALS AND METHODS

**Bioinformatics analysis of *Caldicellulosiruptor* genomes. (i) Genomes and bioinformatics resources for functional annotation.** The following 14 *Caldicellulosiruptor* genomes were downloaded from NCBI Reference Sequence (RefSeq) database: *C. bescii* DSM 6725 (NC_012034), *C. kronotskyensis* 2002 (NC_014720), *C. hydrothermalis* 108 (NC_014652), *C. owensensis* OL (NC_014657), *C. obsidiansis* OB473, *C. danielii* Wai35.B1 (NZ_LACM00000000), *C. acetigenus* DSM 7040 (NZ_ATXO00000000), *C. kristjanssonii* 177R1B (NC_014721), *C. lactoaceticus* 6A (NC_015949), *C. saccharolyticus* DSM 8903 (NC_009437), *Caldicellulosiruptor* sp. F32 (NZ_APGP01000000), *C. changbaiensis* CBS-Z (NZ_CP034791), *C. morganii* Rt8.B8 (NZ_LACO00000000), and *Thermoanaerobacter cellulolyticum* NA10 (renamed *C. naganoensis* NA10, NZ_LACN00000000). All *Caldicellulosiruptor* genomes were functionally annotated using the Rapid Annotation of microbial genomes using Subsystem Technology (RAST) server (49) and were imported to the SEED genomic database to enable comparative genomic analysis and metabolic reconstruction using metabolic subsystems (collections of functionally related proteins) (15). The complete RAST-annotated proteomes with SEED IDs for protein-encoding genes were used for identification orthologs between 14 *Caldicellulosiruptor* strains using a pangenome analysis pipeline based on the identification of bidirectional best hits (50). Genome-wide CAZyme family annotations were assigned using dbCAN server (51), cellular localization of CAZymes was assigned using SignalP and PSORTb (52, 53).

**(ii) Reconstruction of CU metabolic pathways.** For identification of candidate CU genes in the *C. bescii* genome and their functional annotation, we used the previously described bioinformatics workflow (19) and several reference databases including KEGG for reference metabolic pathways and biochemical reactions (54), Pfam for Protein Families and Gene Ontology terms (55), TCDB for Transporter Classification (56), TransportDB for genome-wide transporter prediction (57), CAZy for Carbohydrate Active enZYmes (58). For the majority of candidate CU genes, we also performed protein similarity searches using NCBI BLAST versus UniProt Knowledgebase (UniProtKB), a high-quality curated resource on functional information of experimentally characterized proteins (59), and PaperBLAST, a unique and regularly updated database of proteins described or mentioned in published scientific articles obtained by automated text-mining (60). We further used the subsystem-based comparative genomic approach implemented in the SEED genomic platform and database that capture the taxonomically diverse functional variants of CU enzymes and transporters (15) to analyze the genomic and functional context of CU gene candidates in *C. bescii*. The obtained functional gene annotations are captured in Data Set S1 and summarized in Table 1.

**(iii)** ***Ab initio*** **reconstruction of TF regulons.** For *ab initio* prediction and reconstruction of CU regulons, we utilized the established comparative genomics approach (initially described elsewhere [61] and also reviewed in reference 14) that was previously utilized for *de novo* inference of TF binding site motifs and regulon analysis for numerous TFs implicated in the transcriptional regulation of various CU pathways in diverse taxonomic groups of bacteria (16–19, 21, 32, 37). First, we identified genes encoding potential CU-related TFs by analyzing all representatives of five TF protein families, namely, AraC, DeoR, GntR, LacI, ROK, and RpiR, that are most commonly implicated in CU transcriptional networks in bacteria according to the RegPrecise database (44). For each candidate TF and its orthologs, we analyzed their genomic context across the analyzed *Caldicellulosiruptor* species and, thus excluded putative TFs that do not show the conserved chromosomal clustering with CU genes in the genomes. After the initial assignment of TF candidates to their candidate target genes, we also searched for their previously characterized TF orthologs in the literature (using PaperBLAST [60]) and also in the RegPrecise database, a reference collection of manually curated inferences of TF regulons in bacterial genomes (44) (see Data Set S1). For identification of candidate TFBSs, we collected initial training sets of potentially coregulated CU genes that are located in conserved genomic neighborhoods with a TF gene in the analyzed 14 *Caldicellulosiruptor* genomes. For each training set of genes, we extracted intergenic regions of >50 bp and submitted them to *de novo* DNA motif identification tool, SignalX (62), that uses the expectation-maximization method to find prioritized sets of potential regulatory sites with one of two type of intrinsic symmetry, inverted repeats (palindromes), or tandem repeats. For palindromes, the DNA site size was between 20 and 24 bp, while for inverted repeats we searched for two repeated units, 7 to 11 bp each, that are mutually located with 21- or 32-bp periodicity, corresponding to two or three DNA turns, respectively. For each candidate TFBS motif, we constructed a positional weight matrix (PWM), as previously described (14). For analysis of the Rex regulon, we collected the upstream regions of known Rex-regulated genes in *C. bescii* and their orthologs from other *Caldicellulosiruptor* genomes and applied the motif recognition tool with default parameters to identify a 20-bp palindromic motif. For CggR regulon, its DNA-binding motif was identified as a long-inverted repeat containing two 7-bp half-sites separated by a 23-bp spacer, which is according to the previously reported DNA-binding motifs of CggR from the *Bacillus* and *Lactobacillus* species (37, 38).

The obtained PWM motifs were further used to scan each *Caldicellulosiruptor* genome encoding a TF ortholog via the Genome Explorer software enabling the comparative analysis of identified candidate TF regulon members (62). During the PWM scan, we analyzed the whole-genome upstream regions (up to −300 bp relative to the translation start site) and used thresholds for site scores defined as the lowest score observed in the training set. All identified candidate TFBSs were further validated by phylogenetic footprinting using multiple sequence alignment of upstream gene regions for each group of orthologs, as previously described (16, 17). Multiple alignments of DNA upstream regions (see Fig. S1) were obtained by ClustalW2 integrated in the SEED platform (49). WebLogo (version 2.8.2) was used to draw consensus sequences for derived TFBS motifs (63).

**Construction of *msmK* knockout strain.** Generation of *C. bescii* MACB1080 (Δ*pyrE* Δ*msmK*) was accomplished with allelic exchange in the MACB1018 background (Δ*pyrE*), as described previously (64). Plasmid pGR027, Athe_1803 (*msmK*) deletion construct (see Fig. S2), was generated from PCR products with NEBuilder HiFi DNA Assembly (New England Biolabs, Ipswich, MA). PCR products for construct assembly were generated with PrimeSTAR HS DNA polymerase (TaKaRa Bio, Kusatsu, Shiga, Japan). PCR products for screening were generated with SpeedSTAR HS DNA polymerase (TaKaRa Bio). Plasmid was transformed into NEB 10-beta Competent *E. coli* by heat shock at 42°C. PCR products, plasmid DNA, and genomic DNA were purified with kits from Stratagene and Zymo Research. Prior to transformation into *C. bescii*, pGR027 was methylated with M.CbeI, as described previously (64); methylation was confirmed by lack of digestion in the presence of HaeIII restriction endonuclease. *C. bescii* competent cells were prepared as described previously and transformed by electroporation with a Bio-Rad Gene Pulser (65). Electroporation was performed using 1 μg of methylated plasmid DNA with voltage of 1.8 kV, resistance of 400 Ω, and capacitance of 25 μF. After electroporation, the cells were promptly transferred to recovery media, preheated to 70°C. At 0, 45, 90, and 180 min postelectroporation, 5-ml samples were transferred to 50 ml of selective medium containing 50 μg/ml kanamycin and incubated at 75°C with shaking at 150 rpm until growth was observed (24 to 72 h). Cultures with growth were screened for plasmid integration by PCR and colony purified by three rounds of plating and selection to eliminate merodiploidy before counterselection for second crossover (plasmid loss). Counterselection was accomplished on solid medium using modified DSM 516 medium with 8 mM 5-floroorotic acid (5-FOA) and 20 μM uracil. Clean deletion of Athe_1803 was verified by Sanger sequencing (GeneWiz, South Plainfield, NJ) (see Fig. S2).

**Phenotypic characterization of *msmK* knockout strain.** Unless otherwise specified, *C. bescii* was cultured under anaerobic conditions (80% [vol/vol] nitrogen and 20% [vol/vol] carbon dioxide headspace) at 75°C in modified DSM 516 medium with plate shaking at 150 rpm, as described previously (65). To overcome uracil auxotorophy in modified strains, all media contained 20 μM uracil. Media either contained soluble substrates at a concentration of 25 mM or insoluble substrate loadings of 5 g/liter. Xylose, crystalline cellulose (Avicel), pectin from apple, and beechwood xylan were obtained from Sigma (St. Louis, MO). Glucose was obtained from Amresco (Solon, OH). Cellobiose was obtained from MP Biomedicals (Santa Ana, CA). Prior to medium preparation, pectin and xylan were washed to remove contaminating soluble sugars that could interfere with growth experiments. Pectin and xylan were suspended in cold 70% ethanol, soaked for 30 min, and pelleted by centrifugation at 10,000 × *g* for 10 min. The wash procedure was repeated three times before substrates were dried overnight under airflow in a fume hood. For colony purification during strain construction, solid growth medium and cultures were

prepared using modified DSM 516 medium with 3% agar, as described previously (65). Solid medium cultures were grown at 70°C for 48 h under argon atmosphere.

Growth studies were performed for up to 30 h in biological triplicate in 100-ml serum vials with 50 ml of culture sealed with butyl rubber stoppers. Before beginning growth experiments, MACB1018 (ΔpyrE) and MACB1080 (ΔpyrE ΔrsmK) were revived from glycerol stocks in modified DSM 516 medium with fructose and passaged once before inoculating experimental cultures containing varied sugar substrates. Experimental cultures were inoculated to a cell density of $3.3 \times 10^6$ cells/ml. C. bescii competent cells of MACB1018 (ΔpyrE) were grown to a final optical density at 680 nm of 0.06 to 0.07 at 70°C in 1-liter medium bottles containing 500 ml of low-osmolarity defined medium with amino acids mix (LOD-AA) under argon atmosphere and sealed with butyl rubber stoppers, as described previously (65, 66). Cells were counted on a Petroff-Hausser counting chamber (Hausser Scientific, Horsham, PA). Cultures were counted undiluted or diluted either 5-fold or 10-fold in $1\times$ C. bescii base salts, depending on culture density. For protein estimations, the Bradford method was utilized in 96-well plate format with bovine serum albumin standards (Bio-Rad, Hercules, CA).

**Transcriptomics. (i) Cell growth and harvest.** Complex media for the cultivation of C. bescii DSM 6725, containing 0.5 g liter$^{-1}$ yeast extract, was prepared as described previously (43). Five different growth substrates (glucose, cellobiose, crystalline cellulose [Avicel], xylose, and birchwood xylan; 5 g liter$^{-1}$ [final concentration]) were picked to represent five different conditions for differential transcriptomic analyses. Growth experiments were performed at 78°C as closed anoxic cultures (400-ml volume, shaken at 150 rpm) as described previously (7). Cells were harvested in the mid- to late-exponential-growth phase ($0.5 \times 10^8$ to $1.5 \times 10^8$ cells ml$^{-1}$) by immediately (<1 min) bringing the culture to room temperature through pumping through a glass cooling coil bathed in an ice-water slurry, as previously described (67), a procedure which also led to the removal of most of the insoluble substrate. Cells were separated from the supernatant by centrifugation at $6,000 \times g$ for 5 min and immediately shock-frozen in liquid nitrogen.

**(ii) RNA isolation.** Frozen cell pellets collected from 200 ml samples drawn from each fermenter were resuspended in 2 ml of TRIzol (Invitrogen, CA) and lysed by sonication at 40% power (9 W) on a Misonix 3000 sonicator (Farmingdale, NY). Samples were sonicated for a total of 15 s ($3 \times 5$ s pulse, followed by 1 min off). Chloroform was added, and the sample was mixed and centrifuged. The aqueous layer was removed and mixed 1:1 with 80% ethanol. The mixture was then processed on an RNeasy column (Qiagen, Hilden, Germany) according to the manufacturer's protocol and the on-column DNase digestion. RNA was eluted off the column in 35 μl of RNase-free H$_2$O. The RNA was quantified using a NanoDrop 1000 instrument (Thermo Scientific, Waltham, MA), and the RNA quality was assessed using an Agilent 2100 bioanalyzer (Agilent Technologies, Santa Clara, CA) and a Nano6000 chip kit. Purified total RNA was depleted of rRNA using a Ribo-Zero rRNA removal kit for Gram-positive bacteria (Epicentre-Illumina, Madison, WI) according to the manufacturer's protocol. The depleted sample was concentrated on an RNA Clean & Concentrator-5 (Zymo Research, Irvine, CA) according to the manufacturer's protocol.

**(iii) RNA-Seq library preparation.** Depleted RNA was used to prepare RNA seq libraries using the Epicentre ScriptSeq mRNA-Seq Library preparation kit (Epicentre-Illumina) following the manufacturer's protocol. Agencount AMPure beads (Beckman Coulter, Indianapolis, IN) were used to purify the cDNA, and unique indexes were added during 13 cycles of library amplification. The final RNA-Seq libraries were purified with Agencount AMPure beads (Beckman Coulter) and quantified with a Qubit fluorometer (Life Technologies, Carlsbad, CA). The library quality was assessed on an Agilent 2100 Bioanalyzer (Agilent) using a DNA 7500 DNA Chip (Agilent). Samples were pooled and diluted, and two paired end sequencing runs ($2 \times 76$ bp) were completed on an Illumina MiSeq instrument (Illumina) using the standard protocol and V3 chemistry.

**(iv) RNA-Seq data analysis.** The processing of raw reads was performed based on our previous workflow (68). The raw reads were first assessed for sequencing quality using SolexaQA++ tool kits (69). We performed quality trimming of the raw reads using BWA dynamic trimming algorithm in the SolexaQA++ tool kits to obtain the high-quality reads (phred score > 25 and length after trimming > 25 bp). The high-quality reads were aligned to the C. bescii genome using a high-performance aligner Stampy software (70, 71) with default parameters to generate Bam files. The raw count table was generated from the bam files using BEDTools (72). The raw count table was then used for statistical analysis to identify differential gene expression using the Voom method (73, 74).

**(v) 5′ RNA-Seq for TSS determination.** Total RNA from C. bescii cultures grown on glucose, cellobiose, Avicel, xylose, and xylan were also used as starting material for TSS enrichment. Briefly, 4 μg of total RNA was treated with Ribo-Zero magnetic kit for bacteria (Epicentre) to remove rRNA, and the rRNA-depleted material was purified using Zymo RNA Clean & Concentrator (Irvine, CA) and eluted with 10 μl of RNase-free water. Depleted RNA was quantified on a NanoDrop 1000 (Thermo Scientific) and visualized on an Agilent 2100 bioanalyzer instrument (Agilent Technologies). The 5′ terminator exonuclease (Epicentre) was used to digest RNAs that contained a 5′-monophosphate end in a reaction at 30°C for 30 min that also contained an RNase inhibitor (New England Biolabs, Ipswich, MA). Samples were subsequently purified using RNAClean SPRI beads (Agencourt, Beverly, MA) at $1.8\times$ and eluted with 18 μl of RNase-free dH$_2$O. The exonuclease-treated RNA was then used in a reaction with RNA 5′-pyrophosphohydrolase (RppH) to remove pyrophosphate (New England Biolabs, Ipswich, MA). The reaction proceeded with RNase inhibitor at 37°C for 60 min, and the samples were purified with Agencourt RNAClean SPRI beads ($1.8\times$) and eluted with 11 μl of RNase-free water. The samples were visualized on an Agilent 2100 bioanalyzer instrument. Samples were then ligated with an Illumina 5′ adaptor (5′-ACACUCUUUCCCUACACGACGCUCUUCCGAUCU-3′) using T4 RNA ligase at 37°C for 60 min, followed by Agencourt RNAClean SPRI bead ($1.8\times$) cleanup. An Illumina

3′ adaptor (5′-GTGACTGGAGTTCAGACGTGTGCTCTTCCGATCTNNNNN*N-3′) was then introduced during first-strand synthesis with SuperScript III (Invitrogen) (25°C for 5 min, 50°C for 50 min, 70°C for 15 min, and holding at 4°C), followed by AMpure SPRI bead (Agencourt) purification (1.8×) and elution in 24 $\mu$l of nuclease-free water. Samples were indexed and enriched by PCR (95°C for 3 min; 15 cycles of 98°C for 20 s, 55°C for 30 s, and 72°C for 30 s; 72°C for 7 min; and then holding at 4°C) with KAPA HiFi. To prepare samples for Illumina sequencing, DNA was separated on a 2% 1× TAE ultrapure agarose gel, and gel slices were cut from 300 to 700 bp and purified with a Qiagen Minelute column. Final libraries were quantified with a Qubit fluorometer (Life Technologies), and the library quality was assessed on a Bioanalyzer DNA 7500 DNA chip (Agilent). Samples were then pooled and diluted. Pooled barcoded libraries were sequenced in one direction for 151 bases on an Illumina MiSeq instrument.

In total, five 5′ RNA-Seq data were generated for *C. bescii* grown on different carbon sources. The 5′TSS-generated raw data sets were quality filtered (Illumina CASAVA 1.8) and further analyzed using Rockhopper version 2.03 (75). High-quality reads were aligned to the *C. bescii* genome using Rockhopper with default parameters, except for the minimum expression level used to determine 5′ UTRs was set to 0.1. The 5′ UTRs were analyzed in the integrated IGV genome browser for each candidate CU operon in *C. bescii* to identify TSS position (relative to the translational start of the first gene in operon) and 5′ UTR sequence (see Data Set S2). The identified 5′TSSs were visualized on multiple alignments of orthologous upstream regions of CU genes (see Fig. S1) to locate their candidate −35 and −10 promoter elements that correspond to consensus sequences TTGACA and TATAAT, respectively.

**Data availability.** All RNA-Seq and 5′ RNA-Seq raw sequencing and processed data are available in the Gene Expression Omnibus (GSE163475).

## SUPPLEMENTAL MATERIAL

Supplemental material is available online only.
**DATA SET S1**, XLSX file, 0.1 MB.
**DATA SET S2**, XLSX file, 0.1 MB.
**DATA SET S3**, XLSX file, 2.1 MB.
**DATA SET S4**, XLSX file, 0.1 MB.
**FIG S1**, PDF file, 0.8 MB.
**FIG S2**, PDF file, 0.2 MB.
**FIG S3**, PDF file, 0.1 MB.

## ACKNOWLEDGMENTS

Conceptualization: D.A.R., R.M.K., Y.Z., and M.W.W.A.; research design: D.A.R.; regulon and pathway reconstruction: I.A.R., D.A.R., V.A.R., and A.A.A.; ortholog identification: K.Z. and Y.Z.; RNA-Seq and TSS data analysis: D.A.R., V.A.R., F.L.P., and I.N.; RNA-Seq and 5′ RNA-Seq experiments: M.B., S.D.B., C.M.W., and D.M.K.; wet lab experiments: G.M.R., J.C.R., R.G.B., T.N.N.T., R.M.K., and M.W.W.A. D.A.R. wrote the manuscript. All authors reviewed the manuscript.

This material is based upon work supported by the U.S. Department of Energy, Office of Science, Office of Biological and Environmental Research, Genomic Science Program under Award Number DE-SC0019391. I.N. is partially supported by the National Institute of General Medical Sciences of the National Institutes of Health (award P20GM125503).

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
