## [Reviewer comments · mSystems]

Transcriptional regulation of plant biomass degradation and carbohydrate utilization genes in the extreme thermophile *Caldicellulosiruptor bescii*

Dmitry Rodionov, Irina Rodionova, Vladimir Rodionov, Aleksandr Arzamasov, Ke Zhang, Gabe Rubinstein, Tania Tanwee, Ryan Bing, James Crosby, Intawat Nookaew, Mirko Basen, Steven Brown, Charlotte Wilson, Dawn Klingeman, Farris Poole, Ying Zhang, Robert Kelly, and Michael Adams

Corresponding Author(s): Dmitry Rodionov, Sanford-Burnham-Prebys Medical Discovery Institute

Review Timeline:

Submission Date:	December 21, 2020
Editorial Decision:	February 24, 2021
Revision Received:	April 26, 2021
Accepted:	May 4, 2021

Editor: Zarath Summers

Reviewer(s): The reviewers have opted to remain anonymous.

Transaction Report:

DOI: <https://doi.org/10.1128/mSystems.01345-20>

February 24, 2021

Dr. Dmitry A Rodionov
Sanford-Burnham-Prebys Medical Discovery Institute
10901 North Torrey Pines Road
La Jolla, CA 92037

Re: mSystems01345-20 (Transcriptional regulation of plant biomass degradation and carbohydrate utilization genes in the extreme thermophile *Caldicellulosiruptor bescii*)

Dear Dr. Dmitry A Rodionov:

Thank you for your submission.

Below you will find the comments of the reviewers.

To submit your modified manuscript, log onto the eJP submission site at <https://msystems.msubmit.net/cgi-bin/main.plex>. If you cannot remember your password, click the "Can't remember your password?" link and follow the instructions on the screen. Go to Author Tasks and click the appropriate manuscript title to begin the resubmission process. The information that you entered when you first submitted the paper will be displayed. Please update the information as necessary. Provide (1) point-by-point responses to the issues raised by the reviewers as file type "Response to Reviewers," not in your cover letter, and (2) a PDF file that indicates the changes from the original submission (by highlighting or underlining the changes) as file type "Marked Up Manuscript - For Review Only."

Due to the SARS-CoV-2 pandemic, our typical 60 day deadline for revisions will not be applied. I hope that you will be able to submit a revised manuscript soon, but want to reassure you that the journal will be flexible in terms of timing, particularly if experimental revisions are needed. When you are ready to resubmit, please know that our staff and Editors are working remotely and handling submissions without delay. If you do not wish to modify the manuscript and prefer to submit it to another journal, please notify me of your decision immediately so that the manuscript may be formally withdrawn from consideration by mSystems.

Sincerely,

Zarath Summers

Editor, mSystems

Journals Department
Reviewer comments:

Reviewer #1 (Comments for the Author):

The manuscript, Transcriptional regulation of plant biomass degradation and carbohydrate utilization genes in the extreme thermophile *Caldicellulosiruptor bescii*, describes a comprehensive analysis of the genes and proteins involved in carbohydrate utilization in the thermophilic bacterium *Caldicellulosiruptor bescii*, with a focus on the mechanisms of transcriptional regulation of these carbohydrate utilization genes.

The amount of data is tremendous, and the manuscript is exceptionally informative, providing a solid foundation for the comprehensive understanding of the metabolism and regulation of carbohydrate utilization in *Caldicellulosiruptor bescii*. The manuscript was also well written and organized, summarizing and assigning the data to specific carbohydrates and degradation pathways. There are only a few comments for consideration.

Line 148: Overall, 15 catabolic enzymes... It is a little confusing with the numbers of GH proteins. There seem to be 56 GH proteins in total. Perhaps the authors can simply state how many have signal peptides for secretion, how many are encoded within the glucan degradation locus, and what kind of subset is considered as catabolic?

Line 196: Could the authors briefly explain how the ArsR-family TF (named ArbA) was predicted to control the arabinoside catabolic gene *abfA*? In other sections such as Cellulose and mannan degradation, the reviewer presumes that the individual enzyme activities in the constructed pathways are predicted by comparative genomics along with sequence homology (described in Lines 154-158, 160-162). There may also be more information in the accompanying paper (ref 31). Similar to ArbA, what is the basis for predicting that it is MosR that controls the MosABC transporter with the MOS phosphorylase MosP and cellobiose 2-epimerase MosE? When assigning regulators to regulons, is it the positions of the genes on the genome (Fig. 3), or does the procedure described in Lines 559-562 lead to identification of a single regulator? Reading line 231, it seems that in the case of *pecR*, it is the position of the gene. In terms of *CggR*, it seems to be based on homology to the protein from *Bacillus*. Please also consider *UxaR*, *PecR2*, *HemR*, *RhaR*, *RhaR2* and *BxgRS*. As there are so many genes under examination, the bases for predictions may be diverse, but it would be assuring if they are stated. Perhaps one sentence in the beginning would suffice stating the bases of assigning regulators to regulons.

Line 198: in other species, are the authors referring to other *Caldicellulosiruptor* species, or bacteria in general?

This might be restricted by journal format, but it would be better if the supplementary tables had titles on the top.

Is it correct that the corresponding genes for all proteins in Figure 1 can be identified by referencing the protein/gene name in Fig. 1 and Columns C and D in Supplementary Data 1? Perhaps this could be stated in the legend of Fig. 1?

In Fig. 3, the genes whose transcription starts sites are indicated without other information, are these all upregulated with a particular substrate? Or are these considered to be involved with the genes in the same set, but are not upregulated? Can anything be written in the legend?

Was there difficulty growing the cells on xylan? The reviewer is referring to Fig. 5C.

Others for consideration

Line 180: Eleven

Line 181: Seventeen

Line 979: row?

Reviewer #2 (Comments for the Author):

The authors performed a comprehensive bioinformatics-aided analysis to construct a transcriptional regulatory network to describe how *C. bescii* can utilize plant-derived carbohydrates at molecular levels. Aiming at this, they constructed carbohydrate utilization (CU) pathways and identified DNA binding motifs for the associated regulons in *C. bescii* using functional annotations and a comparative genomics approach. Furthermore, they performed RNA-seq on *C. bescii* grown on different plant-derived mono-/polysaccharides (i.e., cellulose, cellobiose, glucose, xylan, and xylose) to support their bioinformatics-based predictions. Interestingly, the authors also found that MsmK functions as an ATPase component, involving in oligosaccharide transports (CUT1 family).

Overall, the relevant experimental procedures are sound and well-designed. However, the authors' claims regarding the proposed CU regulons (e.g., XylR and XynR for relevant transporters and catabolic enzymes) should be further supported by additional experimental data (i.e., Chip-Seq or at least qRT-PCR for relevant genes etc.). Moreover, detailed transcriptomic data should be included in the main figure and text (e.g., fold-change in different culture conditions). Otherwise, it is too difficult to assess their proposed CU networks. Please refer to the specific comments below.

Major comments

1. Figure 1 should be supported by additional data (i.e., DEG data from RNA-seq and/or quantitative analysis of relevant genes in cells grown on different sugars).
2. Moreover, the transcriptional levels should be incorporated into the main figure to validate the predicted CU pathways. Simple informatics-guided prediction of putative pathways and/or proteins without any supporting empirical data should be minimized and/or toned down.
3. The authors predicted DNA binding motifs and reconstructed regulons for CU-related regulators based on their bioinformatics analysis (refer to section Genomic reconstruction of carbohydrate utilization pathways and regulons). However, some of the transcriptional levels did not match with their predictions. For example, XylR, a repressor responsive to xylose, is predicted to regulate both *xylA* and *xynUVW*. In case of *xylA*, the transcriptional levels in different substrates appear to be consistent with the prediction, while those in *xynUVW* were higher in avicel and xylan (refer to Data

Set_S3). I recommend additional empirical data should be included regarding this issue.

Minor Comments

1. Please check typographical errors throughout the manuscript. For instance, in P2, L32, "engineering ethanol and other..." should be changed to "engineering, ethanol and other...".
2. Figures 3 and 4 can be combined since they are in the same context.
3. P7, L152 & P24, L534: The authors mention "secreted GHs (13 out of 15) and PLs (3 out of 4)." However, it seems that subcellular localizations of proteins are estimated based on web tools (i.e., signalP and PSORTb). If not experimentally validated here or anywhere else, the phrase should be changed to "putative secreted GHs (13 out of 15) and PLs (3 out of 4)".
4. P15, L331-334: In this section, the function of MsmK in CU was characterized based on the growth phenotypes of Δ msmK strains on different substrates. To further clarify its differential functions, the differential expression levels of MsmK should be included in different conditions.
5. P45, L1030: For growth on xylan, growth was reported as endpoint (28 h) protein concentration. Why was an indirect quantification method used specifically for xylan? Possibly because it is difficult to perform cell counting when xylan is used as a substrate?

'Transcriptional regulation of plant biomass degradation and carbohydrate utilization genes in the extreme thermophile *Caldicellulosiruptor bescii*'

Rodionov et al. (*mSystems*01345-20)

Response to Reviewers' comments

Reviewer #1

Line 148: Overall, 15 catabolic enzymes... It is a little confusing with the numbers of GH proteins. There seem to be 56 GH proteins in total. Perhaps the authors can simply state how many have signal peptides for secretion, how many are encoded within the glucan degradation locus, and what kind of subset is considered as catabolic?

RESPONSE: Thank you for your comment. Indeed, this statement could have been clearer. Thus, we have re-phrased and expanded this statement to better report the number of secreted CAZymes identified (lines 155-159) and their deduced functional assignments in the catabolic network of *C. bescii* (lines 167-171).

Line 196: Could the authors briefly explain how the ArsR-family TF (named ArbA) was predicted to control the arabinoside catabolic gene *abfA*? In other sections such as Cellulose and mannan degradation, the reviewer presumes that the individual enzyme activities in the constructed pathways are predicted by comparative genomics along with sequence homology (described in Lines 154-158, 160-162). There may also be more information in the accompanying paper (ref 31). Similar to ArbA, what is the basis for predicting that it is MosR that controls the MosABC transporter with the MOS phosphorylase MosP and cellobiose 2-epimerase MosE? When assigning regulators to regulons, is it the positions of the genes on the genome (Fig. 3), or does the procedure described in Lines 559-562 lead to identification of a single regulator? Reading line 231, it seems that in the case of *pecR*, it is the position of the gene. In terms of CggR, it seems to be based on homology to the protein from *Bacillus*. Please also consider UxaR, PecR2, HemR, RhaR, RhaR2 and BxgRS. As there are so many genes under examination, the bases for predictions may be diverse, but it would be assuring if they are stated. Perhaps one sentence in the beginning would suffice stating the bases of assigning regulators to regulons.

RESPONSE: In most cases, we assigned a regulator to its candidate target genes based on its genomic context as the majority of local regulators of sugar catabolism cluster on the chromosome with their target genes. This has been shown previously in many comparative genomic studies of bacterial regulatory networks (such as in PMIDs 20836887, 24966856, 26438537, 23209028, 26903998, 28751880). To ensure that the conserved chromosomal clustering approach is clearly described, we have updated the text both in the Results (lines 198-199) and Methods (lines 566-570) sections.

Line 198: in other species, are the authors referring to other *Caldicellulosiruptor* species, or bacteria in general?

RESPONSE: Yes, we do refer to the absence of characterized TF orthologs in any bacterial species, according to similarity searches and constructed phylogenetic trees to validate gene orthology. We have modified this statement to make it clearer.

This might be restricted by journal format, but it would be better if the supplementary tables had titles on the top.

RESPONSE: According to the Instructions to Authors, all legends should not be included in the supplemental files; rather, they should appear at the end of the main manuscript text.

Is it correct that the corresponding genes for all proteins in Figure 1 can be identified by referencing the protein/gene name in Fig. 1 and Columns C and D in Supplementary Data 1? Perhaps this could be stated in the legend of Fig. 1?

RESPONSE: Correct. We have also added an explanatory note to Figure 1 legend.

In Fig. 3, the genes whose transcription starts sites are indicated without other information, are these all upregulated with a particular substrate? Or are these considered to be involved with the genes in the same set, but are not upregulated? Can anything be written in the legend?

RESPONSE: We have added an additional Figure S3 to show all genes from Figures 3 and 4 that were upregulated on a particular substrate, according to RNA-Seq data. New Figure S3 contains the detailed information on log₂ fold-changes for those genes in each of five tested carbon sources. The updated Figure 3 legend now refer to Figure S3.

Was there difficulty growing the cells on xylan? The reviewer is referring to Fig. 5C.

RESPONSE: In Figure 5, both reviewers noted the difference in quantification method between growth on xylan and other carbon substrates. As reviewer #2 surmised, indirect measurement (end-point protein estimation) was used, due to difficulties encountered with physically counting cells when xylan was the growth substrate. While growing the parent strain (MACB1018) on xylan did not present any difficulties, quantifying cell growth by our usual method of counting with a hemocytometer was not possible. Due to the wide range of particle sizes present in the xylan substrate, the authors could not clearly distinguish between *C. bescii* cells and small insoluble xylan particles in the counting chamber. To clarify this, the legend of Figure 5 has been modified.

Others for consideration

Line 180: Eleven

Line 181: Seventeen

Line 979: row?

RESPONSE: All corrected.

Reviewer #2

Overall, the relevant experimental procedures are sound and well-designed. However, the authors' claims regarding the proposed CU regulons (e.g., XylR and XynR for relevant transporters and catabolic enzymes) should be further supported by additional experimental data (i.e., Chip-Seq or at least qRT-PCR for relevant genes etc.). Moreover, detailed transcriptomic data should be included in the main figure and text (e.g., fold-change in different culture conditions). Otherwise, it is too difficult to assess their proposed CU networks. Please refer to the specific comments below.

RESPONSE: Our study focused on inferring system-level understanding of transcriptional networks for carbohydrate metabolism using a combination of the comparative genomics-driven bioinformatics identification of TF binding sites and whole-genome transcriptomics measurement for five major carbon

sources most relevant to plant biomass degradation. More detailed study of individual transcriptional regulons falls outside of the scope of this work and is the subject of ongoing efforts. For example, we have looked at two major xylan regulons, XylR and XynR, and generated knockout strains in each of these TFs, and also cloned their genes to purify and study DNA-binding properties of both proteins *in vitro*. Again, these planned experiments are ongoing and will be submitted for publication as an entire story that leverages this current one. The present manuscript, representing a system-level view of the transcriptional network of *C. bescii*, was used to generate a genome-scale metabolic model of this species, as described in the accompanying manuscript submitted to mSystems (Zhang et al. - *mSystems*01351-20). To address the Reviewer's concern, , we have added the differential expression (DE) data from RNA-Seq experiments to Figure 1 and also improved visualization of DE operons with detailed log2-fold change data presented as a heatmap as a part of a new Figure S3.

Major comments

1. Figure 1 should be supported by additional data (i.e., DEG data from RNA-seq and/or quantitative analysis of relevant genes in cells grown on different sugars).

RESPONSE: The substantially revised Figure 1 now include DEG data from RNA-Seq experiments, and all relevant DE genes are highlighted appropriately to DE sugar substrate. The list of these DE genes and operons from the CU pathways is also provided in Table 3, with an indication of associated TF regulons that were predicted to control these genes. By using these two alternative ways of data presentation, we provide both a comprehensive and a detailed view of the reconstructed regulons and their experimental support by available RNA-Seq data.

2. Moreover, the transcriptional levels should be incorporated into the main figure to validate the predicted CU pathways. Simple informatics-guided prediction of putative pathways and/or proteins without any supporting empirical data should be minimized and/or toned down.

RESPONSE: We added new Figure S3 to visualize differential expression of CU genes on five analyzed carbohydrates. Moreover, Additional Data S1 contains empirical data supporting assigned functional roles from the reconstructed CU pathways. These include: (i) functional annotations of experimentally characterized *C. bescii* proteins and their homologs in other organisms (spreadsheet A), and (ii) summary on transcriptomics data obtained in this and previous studies, both for *C. bescii* CU genes and their orthologs in *C. saccharolyticus*. In the revised manuscript, we added a more detailed legend explaining the content of this data file.

3. The authors predicted DNA binding motifs and reconstructed regulons for CU-related regulators based on their bioinformatics analysis (refer to section Genomic reconstruction of carbohydrate utilization pathways and regulons). However, some of the transcriptional levels did not match with their predictions. For example, XylR, a repressor responsive to xylose, is predicted to regulate both *xylA* and *xynUVW*. In case of *xylA*, the transcriptional levels in different substrates appear to be consistent with the prediction, while those in *xynUVW* were higher in avicel and xylan (refer to Data Set_S3). I recommend additional empirical data should be included regarding this issue.

RESPONSE: We agree with this comment. In the revised version, we present detailed transcription data that overlap with the reconstructed TF regulons, as well as their TF binding sites for target genomic loci (Figure S3). Figure S3 clearly shows that both XylR targets, *xynUVW* and *xylA*, are highly upregulated on

xylan, xylose and avicel, when compared to the lowest expression on cellobiose. Furthermore, xylan and xylose had the largest effect on *xylA* (log₂ fold, L2F = 8.3 and 8.1, respectively) and was moderately upregulated on avicel (L2F = 4.8). For the *xynUVW* operon, xylan and avicel had higher L2F values (9.9 and 8.3) compared to L2F values for xylose (5.2). These results suggest that all three substrates are capable of inducing the XylR regulon, which is in line with xylose-rich composition of xylan. However, the observed induction of the XylR regulon by avicel, and to a lesser extent, by glucose, is controversial. One potential explanation could be that there is a broader set of molecular effectors for XylR, which could potentially respond both to xylose and glucose, like it was previously demonstrated by us for another non-orthologous member of the ROK family of repressors in *Thermotoga maritima*. The XylR regulator of xylan/xyloside utilization pathways in *Thermotoga* spp. had broader effector specificity; in addition to xylose (MEC 0.02 mM), the specific DNA-binding of XylR was also repressed by glucose, but with MEC > 0.2 mM (PMID: 23209028). We are currently pursuing *in vitro* characterization of the *C. bescii* XylR

Minor Comments

1. Please check typographical errors throughout the manuscript. For instance, in P2, L32, "engineering ethanol and other..." should be changed to "engineering, ethanol and other...".

RESPONSE: Done. We also checked the rest of manuscript for other typos that were corrected.

2. Figures 3 and 4 can be combined since they are in the same context.

RESPONSE: We think the main difference between these two figures is that each of them summarizes genomic loci and regulons for catabolic pathways specific to different types of polysaccharides, i.e. xylan and cellulose/glucans, and thus illustrate two different sections in the Results. Thus, we propose to keep them separate to maximize clarity. To this point, we generated new Figure S3 that combines all differentially expressed genes loci from Figure 3 and 4 and further added heatmaps of DE values obtained from RNA-Seq data.

3. P7, L152 & P24, L534: The authors mention "secreted GHs (13 out of 15) and PLs (3 out of 4)." However, it seems that subcellular localizations of proteins are estimated based on web tools (i.e., signalP and PSORTb). If not experimentally validated here or anywhere else, the phrase should be changed to "putative secreted GHs (13 out of 15) and PLs (3 out of 4)".

RESPONSE: We agree with this comment and thus changed this phrase to "...have signal peptides and thus are predicted to function as extracellular enzymes". It should be also noted that many of the secreted CAZymes encoded within the GDL locus have been experimentally validated in previous studies (see literature citations in Table S1A).

4. P15, L331-334: In this section, the function of MsmK in CU was characterized based on the growth phenotypes of Δ msmK strains on different substrates. To further clarify its differential functions, the differential expression levels of MsmK should be included in different conditions.

RESPONSE: In the revised manuscript, we add Figure S3 summarizing RNA-Seq data that includes expression data for the *msmK* gene. According to these data, *msmK* is moderately upregulated on xylan, avicel and cellobiose (log₂ fold = 2.0, 1.8 and 1.5, respectively). We also added a note to the Results describing this observation, which is in line its determined role in the uptake of various oligosaccharides (lines 414-417).

5. P45, L1030: For growth on xylan, growth was reported as endpoint (28 h) protein concentration. Why was an indirect quantification method used specifically for xylan? Possibly because it is difficult to perform cell counting when xylan is used as a substrate?

RESPONSE: In Figure 5, both reviewers noted the difference in quantification method between growth on xylan and other carbon substrates. As Reviewer #2 surmised, indirect measurement (end-point protein estimation) was used due to difficulties encountered with physically counting cells when xylan was the growth substrate. While growing the parent strain (MACB1018) on xylan did not present any difficulties, quantifying cell growth by our usual method of counting with a hemocytometer was not possible. Due to the wide range of particle sizes present in the xylan substrate, we could not clearly distinguish between *C. bescii* cells and small insoluble xylan particles in the counting chamber. To clarify this, the legend of Figure 5 has been modified.

May 4, 2021

Dr. Dmitry A Rodionov
Sanford-Burnham-Prebys Medical Discovery Institute
10901 North Torrey Pines Road
La Jolla, CA 92037

Re: mSystems01345-20R1 (Transcriptional regulation of plant biomass degradation and carbohydrate utilization genes in the extreme thermophile *Caldicellulosiruptor bescii*)

Dear Dr. Dmitry A Rodionov:

Your manuscript has been accepted, and I am forwarding it to the ASM Journals Department for publication. For your reference, ASM Journals' address is given below. Before it can be scheduled for publication, your manuscript will be checked by the mSystems senior production editor, Ellie Ghatineh, to make sure that all elements meet the technical requirements for publication. She will contact you if anything needs to be revised before copyediting and production can begin. Otherwise, you will be notified when your proofs are ready to be viewed.

- Minimum resolution of 1280 x 720
- .mov or .mp4. video format
- Provide video in the highest quality possible, but do not exceed 1080p
- Provide a still/profile picture that is 640 (w) x 720 (h) max

We recognize that the video files can become quite large, and so to avoid quality loss ASM suggests sending the video file via <https://www.wetransfer.com/>. When you have a final version of the video and the still ready to share, please send it to Ellie Ghatineh at eghatineh@asmusa.org.

Sincerely,

Zarath Summers
Editor, mSystems

Journals Department
Fig. S3: Accept
Data Set S2: Accept
Data Set S3: Accept
Data Set S4: Accept
Data Set S1: Accept
Fig. S2: Accept
Fig. S1: Accept